# Finding Optimal Tangent Points for Reducing Distortions of Hard-label Attacks

**Chen Ma** [1]   **Xiangyu Guo** [2]   **Li Chen** [1,*]   **Jun-Hai Yong** [1]   **Yisen Wang** [3,4]

[1] School of Software, BNRist, Tsinghua University, Beijing, China
[2] Department of Computer Science and Engineering, University at Buffalo, Buffalo NY, USA
[3] Key Lab. of Machine Perception, School of Artificial Intelligence, Peking University, Beijing, China
[4] Institute for Artificial Intelligence, Peking University, Beijing, China
machenstar@163.com, xiangyug@buffalo.edu
{chenlee,yongjh}@tsinghua.edu.cn, yisen.wang@pku.edu.cn

## Abstract

One major problem in black-box adversarial attacks is the high query complexity in the hard-label attack setting, where only the top-1 predicted label is available. In this paper, we propose a novel geometric-based approach called Tangent Attack (TA), which identifies an optimal tangent point of a virtual hemisphere located on the decision boundary to reduce the distortion of the attack. Assuming the decision boundary is locally flat, we theoretically prove that the minimum $\ell_2$ distortion can be obtained by reaching the decision boundary along the tangent line passing through such tangent point in each iteration. To improve the robustness of our method, we further propose a generalized method which replaces the hemisphere with a semi-ellipsoid to adapt to curved decision boundaries. Our approach is free of pre-training. Extensive experiments conducted on the ImageNet and CIFAR-10 datasets demonstrate that our approach can consume only a small number of queries to achieve the low-magnitude distortion. The implementation source code is released online at `https://github.com/machanic/TangentAttack`.

## 1   Introduction

Adversarial attacks cause deep neural networks (DNNs) to make incorrect predictions by slightly perturbing benign images during the test time. They can be divided into two main categories on the basis of the amount of information exposed by the target model, namely white-box and black-box attacks. Many white-box attacks [6, 29, 32] have been proposed, and they can compute the gradients w.r.t. the target model's input images to generate adversarial examples with the first-order optimization techniques. In contrast, black-box attacks are more practical because they craft adversarial examples without requiring the target model's gradients.

Over the past years, the community has made considerable efforts in developing black-box attacks, and the proposed methods can be divided into transfer- and query-based attacks. Transfer-based attacks [25, 37, 38] generate adversarial examples by using a white-box attack method against a surrogate model to fool the target model. Although there is no need to query the target model in these attacks, the attack success rate can not be guaranteed, especially in the case of targeted attacks. To achieve satisfactory attack success rates, the query-based attacks use elaborate queries to obtain the feedback of the target model for crafting adversarial examples. In the score-based setting, the query-based attacks [2, 12, 22, 28] estimate approximate gradients by querying the predicted scores of the target model at multiple points. However, in most real-world scenarios, the score-based setting

---

*Corresponding author.

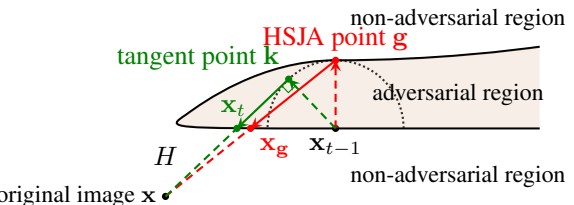

Figure 1: Simplified two-dimensional illustration of our motivation. $H$ is the decision boundary, and $\mathbf{x}_{t-1}$ is the current adversarial example mapped onto the decision boundary at the $(t-1)$-th iteration. HSJA updates $\mathbf{x}_{t-1}$ along the gradient direction to reach $\mathbf{g}$ and then maps it to $H$ at $\mathbf{x_g}$ along the line through $\mathbf{x}$ and $\mathbf{g}$. However, the optimal update should be the tangent point $\mathbf{k}$ because it can be mapped onto $H$ at $\mathbf{x}_t$ that has the shortest distance to the original image $\mathbf{x}$.

is not applicable because the public service returns only the top-1 predicted label (*i.e.,* the hard label) rather than the predicted score. In this case, since the feedback information is limited and the objective function is discontinuous, the attack requires solving a high-dimensional combinatorial optimization problem, which is often challenging.

To reformulate the attack as a real-valued continuous optimization problem, OPT [10], Sign-OPT [11], and RayS [8] focus on minimizing an objective function $g(\theta)$, which is defined as the distance from the original image to the nearest adversarial example along the direction $\theta$. However, when attacking complex models, it may be difficult to find a suitable direction $\theta$ along which adversarial examples exist.

Boundary Attack (BA) [4], HopSkipJumpAttack (HSJA) [7], QEBA [24], qFool [26], and Policy Driven Attack (PDA) [40] eliminate the search of the direction $\theta$. Instead, they start from a large adversarial perturbation and then reduce its distortion while staying in the adversarial region. Because the output labels of the target model flip only near the decision boundary, these attacks restrict their explorations to the regions near the decision boundary. However, they do not thoroughly investigate the geometric properties of the decision boundary to accelerate the attack. For example, PDA uses a reinforcement learning framework to train a policy network to predict search directions, which are not geometrically optimal. In addition, the prediction accuracy of the policy network decreases significantly in the later stages of the iterations, resulting in worse performance of these iterations. HSJA and qFool simply use the gradient $\mathbf{u}$ estimated at the decision boundary as the direction of each update, while ignoring a geometrically critical issue, *i.e.,* $\mathbf{u}$ is not the optimal direction to be followed (Fig. 1). We could explore better search directions at each attack iteration.

To find the optimal search direction for minimizing the distortions of attack, we propose a new geometric-based approach whose motivation is illustrated in Fig. 1. We construct a virtual semicircle $B$ centered at $\mathbf{x}_{t-1}$ to indicate all possible locations that $\mathbf{x}_{t-1}$ can reach along different directions, and the radius of $B$ limits the range of updates for successful attacks. It is easy to observe that moving along the tangent line can reach the nearest location of the decision boundary to the original image $\mathbf{x}$, thereby producing the adversarial example with the minimum distortion. In real attack scenarios, the image data reside in a high-dimensional space, and the semicircle becomes a hemisphere. In this case, the benefit of using tangent points still exists, and we provide the detailed description and the formal proof in Section 3 and appendix.

To summarize, the main contributions of this study are as follows.

1. We cast the problem of minimizing the distortion in hard-label attacks into a geometric problem. We discover that the minimum distortion can be obtained by searching the optimal tangent point of a virtual hemisphere around the adversarial example at each iteration.

2. We propose a novel geometric-based method to obtain a closed-form solution of the optimal tangent point. We provide an intuitive explanation of our approach, as well as a formal proof of its correctness.

3. To improve robustness, we further propose a generalized method that replaces the hemisphere with a semi-ellipsoid to adapt to the target models with curved decision boundaries.

4. Extensive experiments conducted on the CIFAR-10 [23] and ImageNet [14] datasets demonstrate the effectiveness of our approach.

## 2   Related Work

Query-based black-box attacks can be divided into score- and decision-based attack (a.k.a. the hard-label attack). Score-based attacks [1–3, 9, 12, 22, 28, 31] use the predicted probability score to craft adversarial examples, which is not always available in most real-world systems. Hard-label attacks are more useful, but obviously more challenging, because only the top-1 predicted label can be obtained. The hard-label attacks usually fall into three categories.

The first category starts from the original image $\mathbf{x}_0$ and attempts to find a optimal direction $\theta$ to reach the adversarial example. OPT [10] searches an optimal $\theta$ to minimize the distance from $\mathbf{x}_0$ to the nearest adversarial example. Sign-OPT [11] improves the query efficiency of OPT by using a single query to estimate the sign of the directional derivative. RayS [8] eliminates the gradient estimation and proposes a fast check step to efficiently find the direction $\theta$. However, RayS is only applicable to the untargeted attack under the $\ell_\infty$ norm because it is difficult to find a suitable direction to reach the region of the target class in a targeted attack, especially in the case of a large number of classes.

The second category starts from a large perturbation or an image of the target class, and then reduces its distortion while staying in adversarial region, thereby gradually making it closer to the original image. BA [4] and NES [1] are two representative methods, but they have high query complexity. Biased BA [5] reinterprets BA as a biased sampling framework and incorporates different biases to improve the query efficiency. HSJA [7] utilizes the gradient estimation and the binary search to outperform BA. HSJA can be used as the baseline of hard-label attacks. QEBA [24] improves HSJA by using dimension reduction techniques. SurFree [30] and GeoDA [33] improve the performance of HSJA by exploiting geometric properties of DNNs. However, the geometric features of DNNs have not been fully explored, and they do not support the targeted attack. PDA [40] uses a reinforcement learning framework to train a policy network to predict promising directions. However, PDA requires a high-cost pre-training, which is not always available in all tasks.

The third category uses a random sampling technique to improve query efficiency. Customized Adversarial Boundary (CAB) attack [34] uses the current noise to select the sensitive area of images and customize sampling distribution. Evolutionary [16] improves the query efficiency by reducing the dimension of the search space with the stochastic coordinate selection.

The issue of RayS and GeoDA is that they only support untargeted attacks. Because moving the original image far enough in any direction can always make it escape the non-adversarial region in untargeted attacks. However, in targeted attacks, it is difficult to find a suitable direction along which the target class's region exists. In contrast, our approach supports all types of attacks, and we exploit the geometric characteristics of the decision boundaries of DNNs to boost the attack.

## 3   The Proposed Approach

### 3.1   The Goal of Hard-label Attacks

Given a target model $f : \mathbb{R}^d \to \mathbb{R}^k$ and a benign image $\mathbf{x} \in [0, 1]^d$ that is correctly classified using $f$, the goal of the adversary is to slightly perturb $\mathbf{x}$ into $\mathbf{x}_{\mathrm{adv}}$, such that $f(\mathbf{x}_{\mathrm{adv}})$ outputs incorrect prediction. In the hard-label attack, the adversary can only observe the top-1 predicted label of $f$, denoted as $\hat{y} = \arg\max_i f(\mathbf{x}_{\mathrm{adv}})_i$. We define an indicator function $\phi(\cdot)$ of a successful attack:

$$\phi(\mathbf{x}_{\mathrm{adv}}) := \begin{cases} 1 & \text{if } \hat{y} = y_{\mathrm{adv}} \text{ in the targeted attack,} \\ & \text{or } \hat{y} \neq y \text{ in the untargeted attack,} \\ 0 & \text{otherwise,} \end{cases} \tag{1}$$

where $y \in \mathbb{R}$ is the true label of $\mathbf{x}$ and $y_{\mathrm{adv}} \in \mathbb{R}$ is a predefined target class label. In this study, we focus on generating an adversarial example $\mathbf{x}_{\mathrm{adv}}$ that satisfies $\phi(\mathbf{x}_{\mathrm{adv}}) = 1$, such that the distortion $d(\mathbf{x}_{\mathrm{adv}}, \mathbf{x}) := \|\mathbf{x}_{\mathrm{adv}} - \mathbf{x}\|_p$ is minimized. This goal can be formulated as the optimization problem:

$$\min_{\mathbf{x}_{\mathrm{adv}}} d(\mathbf{x}_{\mathrm{adv}}, \mathbf{x}) \quad \text{s.t.} \quad \phi(\mathbf{x}_{\mathrm{adv}}) = 1. \tag{2}$$

### 3.2   Motivation

Most recent attacks belong to the second category (Section 2) follow a common procedure: it starts from an adversarial image yet not close enough to the benign one, then it iteratively searches for a

closer adversarial image. Let us take a typical attack HSJA [7] for example (Fig. 1). First, the attack initializes the adversarial example by using an image of the target class (in a targeted attack) or a noisy version of the original image (in an untargeted attack). Next, it performs the binary search to map the initial sample onto the decision boundary $H$, denoted as $\mathbf{x}_{t-1}$. Then, the algorithm iteratively performs three steps to update $\mathbf{x}_{t-1}$: estimating gradient $\mathbf{u}$ at $H$ by sampling many probes around $\mathbf{x}_{t-1}$; jumping to the point $\mathbf{g}$ along the direction $\mathbf{u}$ with the step size determined by the geometric progression; and mapping $\mathbf{g}$ onto $H$ at $\mathbf{x_g}$ by performing the binary search along the line passing through $\mathbf{g}$ and the original image $\mathbf{x}$. However, geometrically speaking, the estimated gradient $\mathbf{u}$ is not the optimal search direction, and we can find a better boundary point $\mathbf{x}_t$ by connecting $\mathbf{x}$ to a certain point on the semicircle $B$ and then taking the intersection point on $H$. Fig. 1 shows that the tangent point $\mathbf{k}$ is the optimum update because it leads to the minimum distortion of the attack.

### 3.3 Definition of Optimal Tangent Points

There is some experimental evidence that the decision boundaries of DNNs are smooth surfaces with the low curvature [17, 33]. Based on this observation, we approximate the local decision boundary with a hyperplane $H$. Fig. 3 illustrates our problem in three-dimensional space. We will derive our algorithm from $\mathbb{R}^3$ and show that it can be directly extended to higher-dimensional spaces. Lastly, we address the case of curved decision boundaries.

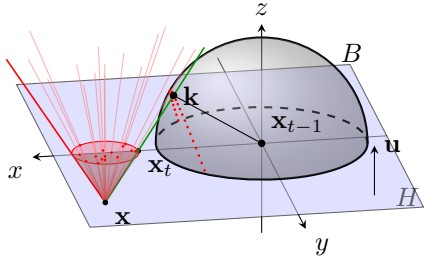

Figure 2: Geometrical explanation of Theorem 1. All points on $H$ that are closer to $\mathbf{x}$ than $\mathbf{x}_t$ are within a red disk, which is the intersection of $H$ and the red cone whose vertex is $\mathbf{x}$. Clearly $\mathbf{k}$ is the only intersection point of the cone and the hemisphere $B$. Thus, of all the lines intersecting $B$, only the intersection of the tangent line and $H$ is closest to $\mathbf{x}$.

As described in Section 3.2, $\mathbf{x}_{t-1}$ denotes the *boundary sample* that already lies on the decision boundary. We create a virtual hemisphere $B$ centered at $\mathbf{x}_{t-1}$ with a radius $R$, which is an estimation of the safe region where we can search for adversarial examples. In the targeted attack, $B$ represents the current estimation of the target class region around $\mathbf{x}_{t-1}$. Ideally, $B$ is small enough to be fully contained in the target class's region, *i.e.,* $\phi(\mathbf{x}') = 1$ for $\forall \mathbf{x}' \in B$.

Then, we need to find a point on $B$ which would produce the minimum distortion when mapped to $H$. In two-dimensional case, this point is the tangent point. However, in $n$-dimensional space where $n \geq 3$, there are infinitely many tangent lines of $B$ passing through $\mathbf{x}$ which create infinitely many tangent points on $B$, shown as the red points in Figs. 2 and 3. Still, we will show that exactly one tangent point can lead to the minimum distortion when mapping it onto $H$ along the tangent line.

Formally, let $\mathbf{k}$ be any point on the surface of $B$, $\mathbf{u}$ be the approximate gradient of $H$ estimated at $\mathbf{x}_{t-1}$, and $\mathbf{x}_t$ be the intersection of $H$ and the line passing through $\mathbf{x}$ and $\mathbf{k}$, we have the following theorem.

**Theorem 1** *Let $H$, $\mathbf{u}$, $\mathbf{k}$, $\mathbf{x}$, and $\mathbf{x}_{t-1}$ be defined above, then the distance $\|\mathbf{x} - \mathbf{x}_t\|$ is minimized if $\mathbf{k}$ is the optimal solution of the following constrained optimization problem:*

$$\arg\max_{\mathbf{k}} \quad \langle \mathbf{k} - \mathbf{x}_{t-1}, \mathbf{u} \rangle \tag{3}$$

$$\text{s.t.} \quad \langle \mathbf{k} - \mathbf{x}_{t-1}, \mathbf{x} - \mathbf{k} \rangle = 0, \tag{4}$$

$$\|\mathbf{k} - \mathbf{x}_{t-1}\| = R, \tag{5}$$

$$\langle \mathbf{k} - \mathbf{x}_{t-1}, \mathbf{u} \rangle \geq 0. \tag{6}$$

*In particular, the optimal $\mathbf{k}$ is in the plane spanned by $\mathbf{u}$ and $\mathbf{x} - \mathbf{x}_{t-1}$.*

The objective function of Eq. (3) is to maximize the projection of the vector $\mathbf{k} - \mathbf{x}_{t-1}$ onto $\mathbf{u}$, which is equivalent to finding $\mathbf{k}$ that is farthest away from $H$. The first constraint ensures that $\mathbf{k}$ is a tangent point. The second constraint indicates $\mathbf{k}$ is on the surface of $B$. The last constraint states that $\mathbf{k}$ cannot appear on the same side of $H$ as $\mathbf{x}$, which is always satisfied if $\|\Pi_H(\mathbf{x} - \mathbf{x}_{t-1})\| \geq R$, where the notation $\Pi_H : \mathbb{R}^n \mapsto H$ denotes the orthogonal projection from $\mathbb{R}^n$ onto the hyperplane $H$. If there is no feasible solution, then our algorithm (Algorithm 1) reduces the radius $R$ to guarantee that the last constraint is always satisfied. The formal proof of Theorem 1 is presented in the appendix. Here,

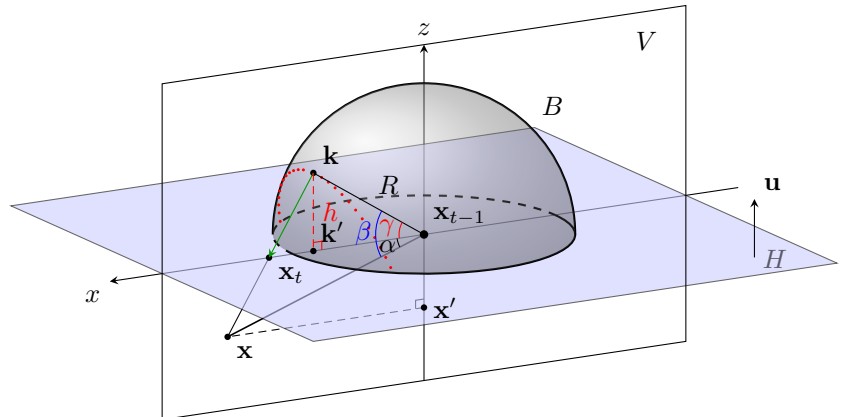

Figure 3: The illustration of the optimal tangent point $\mathbf{k}$ for the flat decision boundary. The tangent point $\mathbf{k}$ is on the surface of a virtual hemisphere $B$ with a radius $R$ centered at the adversarial example $\mathbf{x}_{t-1}$. $\mathbf{x}$ is the original image, $\mathbf{x}_t$ is the intersection point of the tangent line and the decision hyperplane $H$, $\mathbf{k}'$ and $\mathbf{x}'$ are the orthogonal projections of $\mathbf{k}$ and $\mathbf{x}$ onto $H$ and $z$ axis, respectively.

we refer the readers to Fig. 2 for an intuitive geometrical explanation. Eq. (3) is computationally expensive to solve in the high-dimensional space. Fortunately, we show there actually exists a closed-form solution.

### 3.4 Closed-Form Solution of the Optimal Tangent Point

The main intuition of the derivation is illustrated in Fig. 3, which shows an example in $\mathbb{R}^3$. For ease of presentation, we move $\mathbf{x}_{t-1}$ to $\mathbf{0}$. The known variables are $\mathbf{x}$, $\mathbf{x}_{t-1}$, the unit normal vector $\mathbf{u}$ of hyperplane $H$, and the radius $R$. We need to solve for the unknown $\mathbf{k}$. Let $V = \mathrm{span}(\{\mathbf{x}, \mathbf{u}\})$ be the plane spanned by $\mathbf{x}$ and $\mathbf{u}$. According to Theorem 1, we know $\mathbf{k} \in V$. Let us denote the angle between $\mathbf{x}$ and $H$ as $\alpha$, the angle between $\mathbf{x}$ and $\mathbf{k}$ as $\beta$, and the angle between $\mathbf{k}$ and $H$ as $\gamma$. Then, $\beta = \alpha + \gamma$ because all three points $\mathbf{x}$, $\mathbf{k}$, and $\mathbf{x}_{t-1}$ are on the same plane $V$. Because the angle between $\mathbf{x}$ and $\mathbf{u}$ is $\pi \, / \, 2 + \alpha$, we have

$$\langle \mathbf{x}, \mathbf{u} \rangle = \|\mathbf{x}\| \cdot \|\mathbf{u}\| \cdot \cos(\frac{\pi}{2} + \alpha) = \|\mathbf{x}\| \cdot \|\mathbf{u}\| \cdot (-\sin\alpha). \tag{7}$$

Then we have $\sin\alpha = -\frac{\langle \mathbf{x}, \mathbf{u} \rangle}{\|\mathbf{x}\| \cdot \|\mathbf{u}\|}$ and $\cos\alpha = \sqrt{1 - \sin^2\alpha} = \frac{\sqrt{\|\mathbf{x}\|^2 \cdot \|\mathbf{u}\|^2 - \langle \mathbf{x}, \mathbf{u} \rangle^2}}{\|\mathbf{x}\| \cdot \|\mathbf{u}\|}$. By the constraint (4), $\mathbf{x} - \mathbf{k}$ is orthogonal to $\mathbf{k}$. Thus, we have $\cos\beta = R \, / \, \|\mathbf{x}\|$ and $\sin\beta = \sqrt{1 - \cos^2\beta} = \sqrt{\|\mathbf{x}\|^2 - R^2} \, / \, \|\mathbf{x}\|$. Then $\sin\gamma$ and $\cos\gamma$ can be derived as functions of $\alpha$ and $\beta$ from basic facts of trigonometric functions:

$$\begin{aligned} \sin\gamma &= \sin(\beta - \alpha) = \sin\beta\cos\alpha - \cos\beta\sin\alpha, \\ \cos\gamma &= \cos(\beta - \alpha) = \cos\beta\cos\alpha + \sin\beta\sin\alpha. \end{aligned} \tag{8}$$

Now, let $\mathbf{k}' \in H$ be the orthogonal projection of $\mathbf{k}$ onto the plane $H$. The distance between $\mathbf{k}$ and $\mathbf{k}'$ is denoted as $h$ (Fig. 3). Then $h = R \cdot \sin\gamma = R \cdot (\sin\beta\cos\alpha - \cos\beta\sin\alpha)$.

To derive $\mathbf{k}'$, let us denote $\mathbf{x}'$ as the orthogonal projection of $\mathbf{x}$ onto $z$ axis. So we have $\mathbf{x}' = \langle \mathbf{x}, -\mathbf{u} \rangle \cdot (-\mathbf{u}) \, / \, \|\mathbf{u}\|^2 = \langle \mathbf{x}, \mathbf{u} \rangle \cdot \mathbf{u} \, / \, \|\mathbf{u}\|^2$. Then, because $\mathbf{k}'$ and $\mathbf{x} - \mathbf{x}'$ are on the same direction, we have

$$\frac{\mathbf{k}'}{\|\mathbf{k}'\|} = \frac{\mathbf{x} - \mathbf{x}'}{\|\mathbf{x} - \mathbf{x}'\|}. \tag{9}$$

Now, $\|\mathbf{k}'\| = R \cdot \cos\gamma$ and $\mathbf{x}' = \langle \mathbf{x}, \mathbf{u} \rangle \cdot \mathbf{u} \, / \, \|\mathbf{u}\|^2$ are plugged into Eq. (9), and $\mathbf{k}'$ is obtained as

$$\mathbf{k}' = \frac{\mathbf{x} - \mathbf{x}'}{\|\mathbf{x} - \mathbf{x}'\|} \cdot \|\mathbf{k}'\| = \frac{\mathbf{x} - \langle \mathbf{x}, \mathbf{u} \rangle \cdot \mathbf{u} \, / \, \|\mathbf{u}\|^2}{\|\mathbf{x} - \langle \mathbf{x}, \mathbf{u} \rangle \cdot \mathbf{u} \, / \, \|\mathbf{u}\|^2\|} \cdot R \cdot \cos\gamma. \tag{10}$$

Therefore, $\mathbf{k}$ can be derived as

$$\mathbf{k} = \mathbf{k}' + h \cdot \mathbf{u} = \frac{\mathbf{x} - \langle \mathbf{x}, \mathbf{u} \rangle \cdot \mathbf{u} \, / \, \|\mathbf{u}\|^2}{\|\mathbf{x} - \langle \mathbf{x}, \mathbf{u} \rangle \cdot \mathbf{u} \, / \, \|\mathbf{u}\|^2\|} \cdot R \cdot \cos\gamma + h \cdot \mathbf{u}. \tag{11}$$

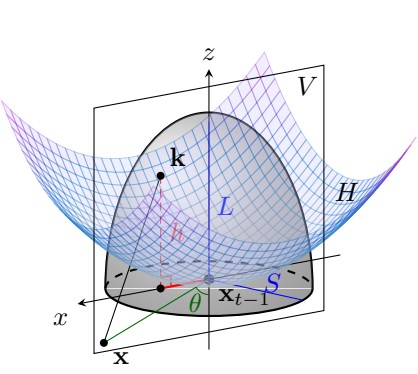

(a) 3D view of Generalized Tangent Attack

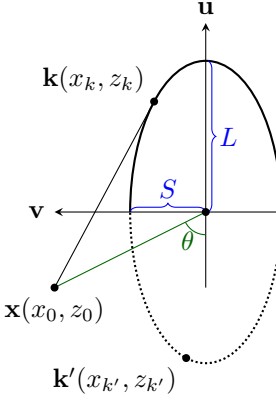

(b) 2D plane $V$ spanned by $\mathbf{x}$ and $\mathbf{u}$.

Figure 4: Illustration of the derivation of Generalized Tangent Attack, which replaces the hemisphere $B$ with a semi-ellipsoid to increase the height of $\mathbf{k}$ to adapt to curved decision boundaries.

Finally, because $\mathbf{x}_{t-1}$ has been moved to the origin, we need to move $\mathbf{k}$ back by adding $\mathbf{x}_{t-1}$.

We remark that although the above derivation is illustrated in $\mathbb{R}^3$, it can be directly applied to higher dimensions. The reason is Theorem 1, which essentially reduces any dimension space to $\mathbb{R}^2$: to find the optimal $\mathbf{k}$, we only need to focus on the plane $V$ spanned by $\mathbf{u}$ and $\mathbf{x} - \mathbf{x}_{t-1}$.

### 3.5 Generalized Tangent Attack

When the local decision boundary is not flat enough, the boundary point obtained via the tangent line may not be the optimal, as shown in Fig. 4a. A simple solution is to simply continue halving the radius $R$ of the hemisphere: as long as $R$ becomes small enough, the local flatness can always be obtained. However, too small a radius will reduce the convergence rate of our algorithm, because the distance between $\mathbf{x}_t$ and $\mathbf{x}_{t-1}$ is proportional to $R$. Therefore, when the classification decision boundary is a curved surface, the attack algorithm should change the shape of hemisphere rather than simply reducing $R$. Based on this idea, we propose the Generalized Tangent Attack (G-TA).

First, although the shape of the decision boundary can be very complex in a high-dimensional space, the important thing for our algorithm is only the situation in the two-dimensional plane spanned by $\mathbf{x}$ and $\mathbf{u}$. In particular, if the decision boundary is "downward" curved (as opposed to the example in Fig. 4a), then searching along the tangent line is still a better approach than HSJA's solution. Thus, the only "bad case" we have to deal with is when the decision boundary is "upward" curved and has a large curvature, as shown in Fig. 4a.

According to Theorem 1, we only need to focus on the plane $V$ spanned by $\mathbf{x}$ and $\mathbf{u}$, as shown in Fig. 4b. Now, $\mathbf{u}$ and $\mathbf{v} := (\mathbf{x} - \langle \mathbf{x}, \mathbf{u} \rangle \cdot \mathbf{u} \,/\, \|\mathbf{u}\|^2) \,/\, \|\mathbf{x} - \langle \mathbf{x}, \mathbf{u} \rangle \cdot \mathbf{u} \,/\, \|\mathbf{u}\|^2\|$ form an orthogonal basis of the plane $V$, then $\mathbf{x}$ can be identified with coordinates $(x_0, z_0)$, i.e., $\mathbf{x} = x_0 \mathbf{v} + z_0 \mathbf{u}$. Let $\theta$ denote the angle between the vector $\mathbf{x}$ and the vector $-\mathbf{u}$, i.e., $\theta = \arccos\left( \frac{\langle \mathbf{x}, -\mathbf{u} \rangle}{\|\mathbf{x}\| \cdot \|\mathbf{u}\|} \right)$. Then $(x_0, z_0) = (\|\mathbf{x}\| \cdot \sin\theta, -\|\mathbf{x}\| \cdot \cos\theta)$. Consider the projection of the ellipsoid on $V$ (which is an ellipse), we denote $L$ as its radius along the direction of $\mathbf{u}$, and $S$ as its radius along the direction of $\mathbf{v}$. Because the optimal tangent point $\mathbf{k}$ lies in the plane $V$, $\mathbf{k}$ can also be identified as $\mathbf{k} = x_k \mathbf{v} + z_k \mathbf{u}$, and we only need to solve for the unknown $(x_k, z_k)$.

The ellipse is characterized by the equation $x^2 \,/\, S^2 + z^2 \,/\, L^2 = 1$, thus the tangent point $\mathbf{k}$ satisfies $x_k^2 \,/\, S^2 + z_k^2 \,/\, L^2 = 1$. Now we view $z$ as a function of $x$, and take the derivative w.r.t. $x$ at both sides of the equation to get the following formula:

$$\frac{2x_k}{S^2} + \frac{2z_k}{L^2} \cdot \left.\frac{dz}{dx}\right|_{x=x_k} = 0. \tag{12}$$

Thus, we have the slope of tangent line at $\mathbf{k}$ be $\left.\frac{dz}{dx}\right|_{x=x_k} = -\frac{x_k L^2}{z_k S^2}$. Therefore, the tangent line can be written as $z - z_k = -\frac{x_k L^2}{z_k S^2}(x - x_k)$. Since the tangent line passes through $\mathbf{x}$, we know

$z_0 - z_k = -\frac{x_k L^2}{z_k S^2}(x_0 - x_k)$. In summary, we can obtain the following system of equations:

$$\begin{cases} L^2 x_k^2 + S^2 z_k^2 - x_k x_0 L^2 - z_0 z_k S^2 = 0, \\ \dfrac{x_k^2}{S^2} + \dfrac{z_k^2}{L^2} = 1. \end{cases} \tag{13}$$

In Eq. (13), the known variables are $L$, $S$, $x_0$ and $z_0$, and the unknown variables that we need to solve for are $x_k$ and $z_k$. In general, there are two solutions for Eq. (13), *i.e.,* $\mathbf{k}$ and $\mathbf{k}'$ depicted in Fig. 4b. Apparently, the solution of the optimal tangent point should satisfy $z_k > 0$, so the one of two solutions in which $z_k > 0$ should be picked:

$$\begin{cases} x_k = \dfrac{S^2 \left( L^2 - z_0 \cdot \frac{L^2 S^2 z_0 + L^2 x_0 \sqrt{-L^2 S^2 + L^2 x_0^2 + S^2 z_0^2}}{L^2 x_0^2 + S^2 z_0^2} \right)}{L^2 \cdot x_0}, \\ z_k = \dfrac{L^2 S^2 z_0 + L^2 x_0 \sqrt{-L^2 S^2 + L^2 x_0^2 + S^2 z_0^2}}{L^2 x_0^2 + S^2 z_0^2}. \end{cases} \tag{14}$$

Finally, the optimal tangent point $\mathbf{k}$ is obtained via $\mathbf{k} = |x_k| \cdot \frac{\mathbf{x} - \langle \mathbf{x}, \mathbf{u} \rangle \cdot \mathbf{u} / \|\mathbf{u}\|^2}{\|\mathbf{x} - \langle \mathbf{x}, \mathbf{u} \rangle \cdot \mathbf{u} / \|\mathbf{u}\|^2\|} + z_k \cdot \mathbf{u}$. In the implementation, the value of $L$ is determined in the same way as the radius $R$ in TA (the hemisphere version). So we fix $L = R$, and use a hyperparameter $r = L/S$ to control the value of $S$. Imagine that in the case of $\mathbb{R}^3$, the semi-ellipsoid becomes "slender" by setting $r > 1$, thereby adapting to the decision boundary of curved surface while preserving a relatively large step size.

### 3.6 The Complete Algorithm

TA (the hemisphere version) and G-TA (the semi-ellipsoid version) can be combined into one algorithm process, which is shown in Algorithm 1. It first performs a binary search to map the initial sample $\tilde{\mathbf{x}}_0$ to the decision boundary. Note that the binary search step always maps any $\mathbf{x}_{\text{adv}}$ to the adversarial side of $H$ that satisfies $\phi(\mathbf{x}_{\text{adv}}) = 1$, hence the attack success rate is always 100%. Then, it performs a *for* loop of $T$ iterations to find the adversarial example that is close to $\mathbf{x}$. In the first iteration, we sample $B_0$ probes around the boundary sample to estimate the gradient, which is increased to $B_0 \sqrt{t}$ at the $t$-th iteration. This is because the error of gradient estimation in the later iterations has a greater impact on the attack performance, so using more samples can reduce the estimation error. Then, a *while* loop is performed to determine a reasonable radius $R$ by repeatedly halving the radius until the tangent point $\mathbf{k}$ is in the adversarial region. Finally, Algorithm 1 uses the binary search method to map $\mathbf{k}$ back to the classification decision boundary to end this iteration.

---

**Algorithm 1** Tangent Attack

---

**Input:** benign image $\mathbf{x}$, attack success indicator function $\phi(\cdot)$ defined in Eq. (1), initial batch size $B_0$, iteration $T$, mode $m \in \{\text{semi-ellipsoid, hemisphere}\}$, radius ratio $r$.
Initialize $\tilde{\mathbf{x}}_0$ that satisfies $\phi(\tilde{\mathbf{x}}_0) = 1$;
$\mathbf{x}_0 \leftarrow \text{BinarySearch}(\tilde{\mathbf{x}}_0, \mathbf{x}, \phi)$;                               ▷ boundary search
$d_0 = \|\mathbf{x}_0 - \mathbf{x}\|$;
**for** $t$ **in** $1, \ldots, T$ **do**
    Sample $B_t \leftarrow B_0 \sqrt{t}$ random vectors to estimate the gradient $\mathbf{u}$;
    Initialize $R \leftarrow d_{t-1}/\sqrt{t}$;                               ▷ the initial radius
    **while true do**
        Compute the optimal tangent point $\mathbf{k}$ based on Eq. (11) **if** $m =$ hemisphere **else** Eq. (14);
        $R \leftarrow \frac{R}{2}$;                  ▷ search the radius, and we set $L = R, S = \frac{L}{r}$ if $m =$ semi-ellipsoid
        **if** $\phi(\mathbf{k}) = 1$ **then**
            **break**;
        **end if**
    **end while**
    $\mathbf{k} \leftarrow \text{Clip}(\mathbf{k}, 0, 1)$;
    $\mathbf{x}_t \leftarrow \text{BinarySearch}(\mathbf{k}, \mathbf{x}, \phi)$;                               ▷ boundary search
    $d_t = \|\mathbf{x}_t - \mathbf{x}\|$;
**end for**

---

# 4 Experiment

## 4.1 Experimental Setting

**Datasets.** TA and G-TA are evaluated on two datasets, namely CIFAR-10 and ImageNet with the image resolutions of $32 \times 32 \times 3$ and $299 \times 299 \times 3$, respectively. We randomly select 1,000 correctly classified images from their validation sets for experiments.

**Method Setting.** The initial batch size $B_0$ is set to 100, which means the algorithm samples 100 probes for estimating a gradient at the first iteration. The threshold $\gamma$ that controls the termination of the binary search is set to 1.0 in the CIFAR-10 dataset and 1,000 in the ImageNet dataset. The radius ratio $r$ is set to 1.5 in the CIFAR-10 dataset and 1.1 in the ImageNet dataset. Besides, we also set $r$ to 1.5 when attacking defense models. In targeted attacks, the target class label is set to $y_{adv} = (y + 1) \bmod C$, where $y$ is the true label, and $C$ is the number of classes.

**Compared Methods.** The advantage of our method is that it supports all types of attacks, including both untargeted and targeted attacks under $\ell_2$ norm and $\ell_\infty$ norm constraints. Therefore, for complete and fair comparisons, we select the compared methods that support both untargeted and targeted attacks with state-of-the-art performance, including Boundary Attack (BA) [4], Sign-OPT [11], SVM-OPT [11], and HopSkipJumpAttack (HSJA) [7]. HSJA is adopted as the baseline, whose hyperparameters are set to be the same with ours (*e.g,* the same initial batch size $B_0$ and threshold $\gamma$). In addition, QEBA [24] is a HSJA-based method which has three variants: QEBA-I, QEBA-S and QEBA-F. We select QEBA-S in the additional experiment to verify whether the proposed method can improve attack performance of other HSJA-based method. For the consistency of experiments, we translate the implementations of Sign-OPT, SVM-OPT and HSJA from the official NumPy version into the PyTorch version by replacing each NumPy function with the corresponding PyTorch function. Thus, the two versions behave exactly the same. In the targeted attack, we randomly select an image from the target class of the validation set to be the initial sample of HSJA, BA, TA and G-TA. For fair comparison, we set the initial direction $\theta_0$ of Sign-OPT and SVM-OPT to the direction of a randomly selected image of the target class. The detailed settings are presented in the appendix.

**Target Models.** In the CIFAR-10 dataset, we select four target models implemented using the PyTorch framework[2]: (1) a 272-layer PyramidNet+ShakeDrop network (PyramidNet-272) [19, 39] trained using AutoAugment [13] ; (2) a model obtained through a neural architecture search called GDAS [15]; (3) a wide residual network with 28 layers and 10 times width expansion (WRN-28) [41]; and (4) a wide residual network with 40 layers (WRN-40) [41]. In the ImageNet dataset, we select four target models from an off-the-shelf library containing pre-trained weights[3]: (1) Inception-v3 [36], (2) Inception-v4 [35], (3) ResNet-101 [20], and (4) SENet-154 [21].

**Evaluation Metric.** Following previous studies [40], we report the mean $\ell_2$ distortions as $\frac{1}{|\mathbf{X}|} \sum_{\mathbf{x} \in \mathbf{X}} (\|\mathbf{x}_{adv} - \mathbf{x}\|)$ under different query budgets, where $\mathbf{X}$ is the test set.

## 4.2 Comparisons with State-of-the-Art Methods

**Results of Attacks against Undefended Models.** Tables 1 and 2 show the experimental results on the ImageNet and CIFAR-10 datasets. We derive two conclusions based on the results:

(1) We found that the experiments of CIFAR-10 requires a larger radius ratio $r$ than that of ImageNet to achieve the satisfactory performance of G-TA. We speculate that the reason is that the target models of ImageNet have relatively flat decision boundaries.

(2) TA is more effective in targeted attacks, while the G-TA performs better in untargeted ones. This is because the adversarial region of the target class is narrower and more scattered, making the local classification decision boundary smoother, so that Theorem 1 holds and TA performs better.

In addition, one unique benefit of our approach is that it can be used as a performance enhanced plug-in when combining it with other HSJA-based approaches (*e.g.*, QEBA-S). Specifically, the method is to change the jump directions of boundary samples of QEBA-S to the directions of the optimal tangent points. We present that "QEBA-S + TA" can further improve the performance of QEBA-S, as shown in Table 3.

---

[2]Pre-trained weights: `https://github.com/machanic/SimulatorAttack`
[3]Pre-trained weights: `https://github.com/Cadene/pretrained-models.pytorch`

Table 1: Mean $\ell_2$ distortions of different query budgets on the ImageNet dataset, where $r = 1.1$.

| Target Model | Method | Targeted Attack | | | | | | Untargeted Attack | | | | | |
|---|---|---|---|---|---|---|---|---|---|---|---|---|---|
| | | @300 | @1K | @2K | @5K | @8K | @10K | @300 | @1K | @2K | @5K | @8K | @10K |
| Inception-v3 | BA [4] | 111.798 | 108.044 | 106.283 | 102.715 | 86.931 | 78.326 | - | 107.558 | 102.309 | 95.776 | 78.668 | 60.296 |
| | Sign-OPT [11] | 103.939 | 87.706 | 71.291 | 46.744 | 34.640 | 29.414 | 121.085 | 79.158 | 43.642 | 16.625 | 10.557 | 8.680 |
| | SVM-OPT [11] | **101.630** | 82.950 | 67.965 | 46.275 | 35.694 | 31.106 | 121.135 | 66.027 | 36.763 | 15.736 | 10.501 | 8.789 |
| | HSJA [7] | 111.562 | 95.295 | 82.111 | 52.544 | 37.395 | 30.425 | 103.605 | 57.295 | 37.185 | 15.484 | 9.989 | 7.967 |
| | TA | 103.781 | **80.327** | **66.708** | 42.121 | **30.846** | 25.566 | **94.752** | 52.523 | 35.229 | 15.040 | 9.748 | 7.793 |
| | G-TA | 103.724 | 81.089 | 67.168 | 42.434 | 31.011 | 25.587 | 94.972 | **52.278** | **34.734** | **14.850** | **9.673** | **7.757** |
| Inception-v4 | BA [4] | 110.343 | 106.616 | 104.586 | 100.321 | 84.058 | 75.507 | - | 116.075 | 111.474 | 104.451 | 86.572 | 66.283 |
| | Sign-OPT [11] | 101.620 | 85.731 | 69.719 | 46.416 | 34.957 | 30.004 | 132.991 | 86.431 | 48.292 | 18.678 | 11.567 | 9.262 |
| | SVM-OPT [11] | **99.856** | 81.342 | 66.982 | 45.667 | 35.477 | 31.152 | 132.227 | 72.920 | 41.095 | 17.611 | 11.418 | 9.372 |
| | HSJA [7] | 109.670 | 93.916 | 80.937 | 52.358 | 37.773 | 30.958 | 110.727 | 63.731 | 42.290 | 17.936 | 11.367 | 8.911 |
| | TA | 101.666 | **78.683** | **65.304** | **41.629** | 30.993 | 25.958 | 101.207 | 58.616 | 40.314 | 17.639 | 11.304 | 8.907 |
| | G-TA | 101.495 | 79.210 | 65.888 | 42.002 | **30.965** | **25.847** | 101.324 | 58.718 | **40.106** | 17.296 | **11.032** | **8.691** |
| SENet-154 | BA [4] | 81.090 | 77.723 | 76.122 | 71.967 | 55.953 | 47.652 | - | 75.998 | 71.671 | 66.983 | 53.917 | 40.725 |
| | Sign-OPT [11] | 75.722 | 62.876 | 49.191 | 30.155 | 21.333 | 17.672 | 70.035 | 47.705 | 27.314 | 10.890 | 6.643 | 5.245 |
| | SVM-OPT [11] | 75.680 | 58.932 | 47.073 | 30.348 | 22.553 | 19.312 | 69.854 | 40.291 | **23.692** | 10.494 | 6.666 | 5.409 |
| | HSJA [7] | 77.035 | 63.488 | 51.802 | 30.138 | 19.680 | 16.261 | 71.248 | 38.035 | 24.895 | 10.218 | 5.855 | 4.842 |
| | TA | 70.739 | 55.256 | **43.694** | **24.961** | **16.756** | **13.876** | **65.589** | 35.689 | 24.037 | 10.039 | 5.774 | 4.766 |
| | G-TA | **70.591** | **55.224** | 44.047 | 25.041 | 16.854 | 14.047 | 65.846 | **35.601** | 23.730 | **9.902** | **5.720** | **4.738** |
| ResNet-101 | BA [4] | 81.565 | 77.903 | 76.366 | 72.392 | 58.746 | 51.679 | - | 64.007 | 60.389 | 56.544 | 44.175 | 31.371 |
| | Sign-OPT [11] | 76.732 | 63.939 | 51.231 | 32.439 | 23.160 | 19.248 | 56.244 | 38.282 | 21.985 | 10.048 | 7.050 | 6.050 |
| | SVM-OPT [11] | 77.031 | 61.417 | 49.842 | 32.806 | 24.553 | 20.964 | 55.894 | 32.638 | 19.409 | 9.830 | 7.185 | 6.281 |
| | HSJA [7] | 76.121 | 63.091 | 52.301 | 31.018 | 20.472 | 16.911 | 56.264 | 27.443 | 17.717 | 7.649 | 4.723 | **4.019** |
| | TA | **72.434** | **57.969** | **47.142** | **27.699** | **18.788** | **15.414** | 53.197 | 26.777 | 17.651 | 7.730 | 4.822 | 4.107 |
| | G-TA | 72.459 | 58.320 | 47.297 | 27.905 | 19.045 | 15.633 | **53.142** | **26.597** | **17.345** | **7.568** | **4.712** | 4.021 |

Table 2: Mean $\ell_2$ distortions with different query budgets on the CIFAR-10 dataset, where $r = 1.5$.

| Target Model | Method | Targeted Attack | | | | | | Untargeted Attack | | | | | |
|---|---|---|---|---|---|---|---|---|---|---|---|---|---|
| | | @300 | @1K | @2K | @5K | @8K | @10K | @300 | @1K | @2K | @5K | @8K | @10K |
| PyramidNet-272 | BA [4] | 8.651 | 8.073 | 8.013 | 6.387 | 4.189 | 3.333 | - | 5.636 | 4.725 | 4.414 | 2.750 | 1.696 |
| | Sign-OPT [11] | 8.279 | 6.331 | 4.250 | 1.718 | 0.960 | 0.718 | 4.387 | 2.334 | 1.178 | 0.403 | 0.267 | 0.226 |
| | SVM-OPT [11] | 9.207 | 6.801 | 4.530 | 2.010 | 1.207 | 0.947 | 4.481 | 2.318 | 1.093 | 0.414 | 0.276 | 0.236 |
| | HSJA [7] | 7.917 | 4.329 | 2.523 | **0.793** | **0.489** | **0.397** | 4.505 | 1.279 | 0.713 | 0.333 | 0.255 | 0.227 |
| | TA | 7.943 | **4.267** | 2.488 | 0.809 | 0.503 | 0.406 | **4.256** | 1.275 | 0.710 | **0.329** | 0.253 | 0.226 |
| | G-TA | **7.816** | 4.277 | **2.469** | 0.803 | 0.505 | 0.412 | 4.432 | **1.270** | **0.702** | **0.329** | **0.252** | **0.225** |
| GDAS | BA [4] | 8.487 | 7.885 | 7.821 | 6.034 | 3.632 | 2.703 | - | 2.717 | 2.514 | 2.373 | 1.642 | 1.106 |
| | Sign-OPT [11] | 8.372 | 6.514 | 4.351 | 1.827 | 0.987 | 0.711 | 4.917 | 4.159 | 3.260 | 1.352 | 0.452 | 0.250 |
| | SVM-OPT [11] | 9.529 | 7.243 | 5.092 | 2.347 | 1.317 | 0.958 | 4.909 | 3.950 | 2.736 | 1.082 | 0.371 | 0.234 |
| | HSJA [7] | 7.714 | 3.566 | 1.966 | 0.591 | 0.365 | 0.301 | 2.188 | 0.756 | 0.483 | 0.261 | 0.208 | 0.189 |
| | TA | **7.674** | **3.529** | **1.946** | 0.585 | 0.366 | 0.302 | 2.190 | 0.774 | 0.485 | 0.257 | 0.206 | 0.187 |
| | G-TA | 7.697 | 3.558 | 1.959 | **0.583** | **0.361** | **0.298** | **2.161** | **0.745** | **0.476** | **0.255** | **0.204** | **0.185** |
| WRN-28 | BA [4] | 8.688 | 8.046 | 7.984 | 5.786 | 2.486 | 1.555 | - | 4.425 | 3.648 | 3.435 | 1.543 | 0.832 |
| | Sign-OPT [11] | 8.258 | 5.576 | 3.260 | 1.087 | 0.593 | 0.459 | 3.093 | 1.494 | 0.828 | 0.319 | **0.239** | **0.213** |
| | SVM-OPT [11] | 9.516 | 5.968 | 3.744 | 1.367 | 0.728 | 0.553 | 2.977 | 1.466 | 0.723 | 0.325 | 0.245 | 0.221 |
| | HSJA [7] | 6.810 | 2.603 | 1.326 | 0.518 | 0.389 | 0.347 | 3.052 | 0.797 | 0.508 | 0.299 | 0.250 | 0.232 |
| | TA | 6.802 | 2.556 | 1.311 | 0.519 | 0.394 | 0.353 | **2.974** | 0.785 | 0.496 | 0.293 | 0.249 | 0.233 |
| | G-TA | **6.755** | **2.543** | **1.281** | **0.513** | **0.387** | **0.345** | 2.995 | **0.782** | 0.502 | 0.298 | 0.250 | 0.232 |
| WRN-40 | BA [4] | 8.658 | 8.014 | 7.953 | 5.738 | 2.484 | 1.566 | - | 4.377 | 3.586 | 3.367 | 1.487 | 0.821 |
| | Sign-OPT [11] | 8.156 | 5.579 | 3.300 | 1.186 | 0.646 | 0.501 | 4.754 | 3.239 | 1.885 | 0.311 | **0.226** | **0.201** |
| | SVM-OPT [11] | 9.339 | 6.061 | 3.840 | 1.445 | 0.800 | 0.605 | 4.457 | 2.756 | 0.739 | 0.310 | 0.229 | 0.206 |
| | HSJA [7] | 6.909 | 2.648 | 1.330 | 0.528 | 0.400 | **0.357** | 2.992 | 0.777 | 0.498 | 0.290 | 0.242 | 0.225 |
| | TA | 6.944 | **2.579** | **1.295** | **0.523** | **0.398** | 0.358 | **2.926** | **0.770** | **0.490** | **0.288** | 0.243 | 0.227 |
| | G-TA | **6.783** | 2.605 | 1.320 | 0.535 | 0.403 | 0.361 | 2.952 | 0.772 | 0.492 | **0.288** | 0.241 | 0.223 |

Table 3: Experimental results of the combined method of QEBA-S and TA.

| Target Model | Method | Targeted Attack | | | | | |
|---|---|---|---|---|---|---|
| | | @300 | @1K | @2K | @5K | @8K | @10K |
| Inception-v3 | QEBA-S [24] | **100.295** | 79.604 | 63.621 | 35.194 | 22.773 | 18.414 |
| | QEBA-S + TA | 104.490 | **75.622** | **59.836** | **33.112** | **22.329** | **17.799** |
| Inception-v4 | QEBA-S [24] | **97.772** | 77.347 | 62.451 | 35.275 | 23.204 | 19.002 |
| | QEBA-S + TA | 101.845 | **73.838** | **58.554** | **33.288** | **23.160** | **18.736** |
| SENet-154 | QEBA-S [24] | **72.831** | 55.367 | 42.674 | 21.988 | 13.888 | 11.210 |
| | QEBA-S + TA | 76.547 | **52.269** | **39.740** | **20.608** | **13.873** | **11.016** |
| ResNet-101 | QEBA-S [24] | **75.567** | 57.929 | 44.983 | 23.209 | 14.402 | 11.467 |
| | QEBA-S + TA | 78.709 | **53.917** | **41.245** | **21.198** | **13.856** | **10.773** |

**Results of Attacks against Defense Models.** We conduct the experiments of untargeted attacks on defense models. Fig. 5 shows the experimental results on the CIFAR-10 dataset. We select 4 defense models: (1) a model obtained through adversarial training, abbreviated as AT [29]; (2) an

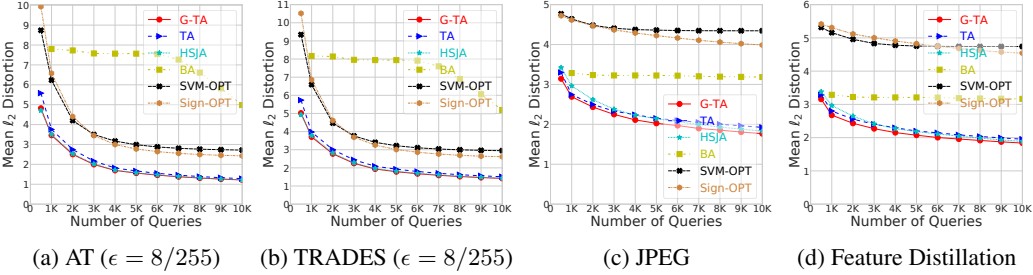

(a) AT ($\epsilon = 8/255$)    (b) TRADES ($\epsilon = 8/255$)    (c) JPEG    (d) Feature Distillation

Figure 5: Experimental results of the attacks against defense models with the backbone of ResNet-50.

improved AT that optimizes a regularized surrogate loss, named as TRADES [42]; (3) an image-transformation-based defense called JPEG [18]; and (4) a DNN-oriented compression defense called Feature Distillation [27]. All defense models adopt ResNet-50 as the backbone. Previous studies [43] have shown that AT and TRADES have the issue of robust overfitting, which leads to a significant increase in the curvature of the classification decision boundary. Figs. 5a and 5b show that G-TA outperforms TA when attacking AT and TRADES. This advantage is also demonstrated in attacking other defenses (*e.g.,* Figs. 5d and 5c), proving the effectiveness of G-TA in attacking defense models.

### 4.3 Comprehensive Understanding of Tangent Attack

In the ablation studies, we conduct the experiments of the targeted attacks on the ImageNet dataset to understand our approach in depth, and the target model is ResNet-50. The results are shown in Fig. 6.

**Initialization.** Our algorithm starts with an image $\tilde{\mathbf{x}}_0$ selected from the target class, and we study three selection strategies: (1) a randomly selected image, (2) the image with the shortest distance to the original image, and (3) the image with the longest distance to the original image. Fig. 6a shows that the strategy of (2) achieves the best performance.

**Radius Ratio.** Fig. 6b shows that the performance of G-TA is not sensitive to the radius ratio $r$.

**Jump Direction.** Fig. 6c shows the effects of different jump directions. RandomHSJA is a variant of HSJA which adopts a random vector $\mathbf{r}$ that satisfies $\langle \mathbf{r}, \mathbf{u} \rangle > 0$ as the jump direction. Fig. 6c verifies the benefit of jumping to the optimal tangent point.

**Initial Batch Size.** In general, Fig. 6d shows that a smaller $B_0$ performs better since it saves queries. But $B_0 = 5$ performs worse than $B_0 = 30$ because it uses too few samples for gradient estimation.

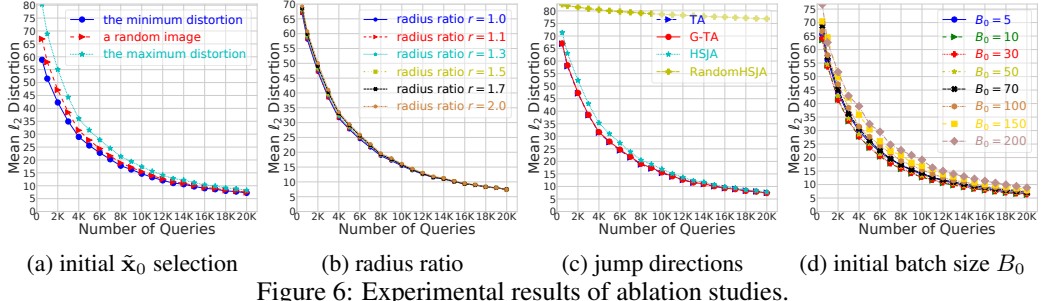

(a) initial $\tilde{\mathbf{x}}_0$ selection    (b) radius ratio    (c) jump directions    (d) initial batch size $B_0$

Figure 6: Experimental results of ablation studies.

## 5 Conclusion

In this paper, we propose a new geometric-based method for query-efficient hard-label black-box attacks. Our method relies on the observation that the minimum $\ell_2$ distortion can be obtained by searching a boundary point along a tangent line of a virtual hemisphere. We offer a closed-form solution for computing the optimal tangent point and provide a formal proof of its correctness. We further propose a generalized method that replaces the hemisphere with a semi-ellipsoid to adapt to the target models with curved decision boundaries. Lastly, we evaluate our approach through extensive experiments and show its superior performance compared with baseline methods.

## Acknowledgments

This research is partially supported by the National Key R&D Program of China (2019YFB1405703) and TC190A4DA/3, the National Natural Science Foundation of China (Grant Nos. 62021002, 61972221). Yisen Wang is partially supported by the National Natural Science Foundation of China under Grant 62006153, and Project 2020BD006 supported by PKU-Baidu Fund.

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
