# Appendix of Finding Optimal Tangent Points for Reducing Distortions of Hard-label Attacks

**Chen Ma** [1] **Xiangyu Guo** [2] **Li Chen** [1,*] **Jun-Hai Yong** [1] **Yisen Wang** [3,4]
[1] School of Software, BNRist, Tsinghua University, Beijing, China
[2] Department of Computer Science and Engineering, University at Buffalo, Buffalo NY, USA
[3] Key Lab. of Machine Perception, School of Artificial Intelligence, Peking University, Beijing, China
[4] Institute for Artificial Intelligence, Peking University, Beijing, China
machenstar@163.com, xiangyug@buffalo.edu
{chenlee,yongjh}@tsinghua.edu.cn, yisen.wang@pku.edu.cn

## A  Potential Negative Societal Impacts

The adversarial attack is a major security concern in the real-world machine learning system, because the generated adversarial perturbation could be used for the malicious purpose. Our study relies only on the top-1 predicted label to craft the adversarial examples which is applicable to most real-world systems, making it more useful and practical. Although the experiments in this paper are about attacking the image classifier, this method can be used in other settings, such as the object detection, the recommender system, the facial recognition system, and the autonomous driving. In summary, this study could be used in harmful ways by malicious users.

In a broader perspective, the adversarial example is not restricted to malicious applications, and it can be used in the positive side, *e.g.,* the generation of CAPTCHA and the privacy protection. In particular, the study of adversarial attacks can promote the defense techniques. In recent years, many proposed defenses are broken by the latest attacks, which stimulates the development of defenses.

Our results also point to the potential defense techniques against hard-label attacks. For example, the defense can prohibit queries near the decision boundary, then the approximate gradient cannot be estimated, making Tangent Attack ineffective. Another possible defense is to add random perturbations to the input image to prevent effective gradient estimation, or predict random classification labels for samples near the classification decision boundary.

## B  Proof of Theorem 1

### B.1  Notations and Assumption

Before we formally prove Theorem 1, let us first define the notations that will be used in the proof. Let $\mathbf{x}$ denote the original image, and w.l.o.g. we assume the boundary sample $\mathbf{x}_{t-1} = \mathbf{0}$ be the origin of the coordinate axis. Let $B$ denote a $n$-dimensional ball centered at $\mathbf{x}_{t-1}$ with the radius of $R$, and its surface is denoted as $S := \partial B$. Note that $B$ denotes a complete ball in this proof. However, $B$ denotes the hemisphere in the main text of the paper. Theorem 1 assumes that the classification decision boundary of the target model is the hyperplane $H$, which is defined by its unit normal vector $\mathbf{u}$. Then, the hyperplane $H$ divides $\mathbb{R}^n$ into two half-spaces:

$$
\begin{aligned}
H_{\geq 0} &= \{\mathbf{v} \in \mathbb{R}^n \mid \langle \mathbf{v}, \mathbf{u} \rangle \geq 0\}, \\
H_{\leq 0} &= \{\mathbf{v} \in \mathbb{R}^n \mid \langle \mathbf{v}, \mathbf{u} \rangle \leq 0\}.
\end{aligned}
\tag{1}
$$

In the attack, $H_{\geq 0}$ mainly contains the adversarial region, and $H_{\leq 0}$ represents the non-adversarial region. In Fig. 1, we visually represent the hyperplane $H$ and two half-spaces in $\mathbb{R}^3$. Suppose

---

*Corresponding author.

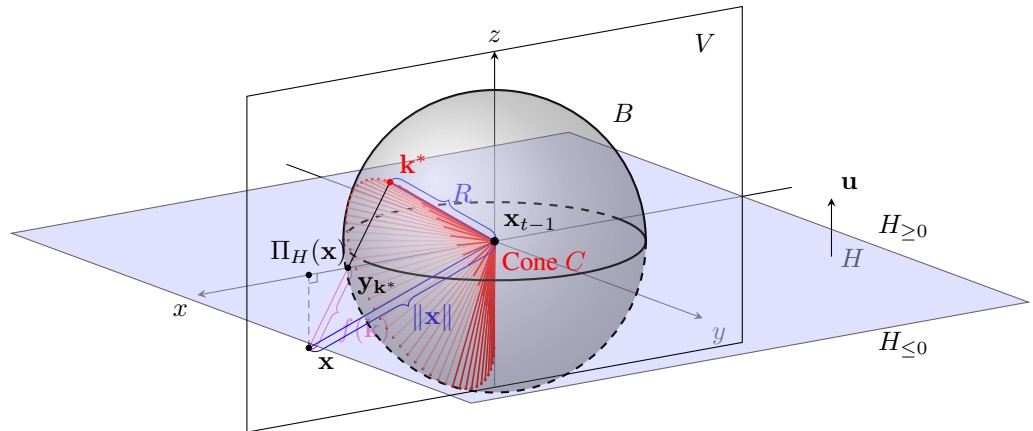

Figure 1: Illustration of the entities defined in the proof, where $C$ is a convex cone whose boundary intersects with the circle formed by all the tangent points from $\mathbf{x}$ to the ball $B$.

$\mathbf{x} \in H_{\leq 0} \setminus B$ is a fixed point outside $B$ such that $\langle \mathbf{x}, \mathbf{u} \rangle < 0$. Now, let us define the cosine function $\cos(\mathbf{a}, \mathbf{b}) := \frac{\langle \mathbf{a}, \mathbf{b} \rangle}{\|\mathbf{a}\|\|\mathbf{b}\|}$ to represent the cosine of the angle between two vectors, then we can define the convex cone $C$ with $\mathbf{x}_{t-1}$ as its vertex, as shown below:

$$C := \left\{ \mathbf{v} \in \mathbb{R}^n \mid \cos(\mathbf{v}, \mathbf{x}) \geq \frac{R}{\|\mathbf{x}\|} \right\}. \tag{2}$$

Fig. 1 demonstrates the convex cone $C$ in $\mathbb{R}^3$. For $\mathbf{v} \in S \cap C$ that satisfies $\cos(\mathbf{v}, \mathbf{x}) = R \,/\, \|\mathbf{x}\|$, the equation $\|\mathbf{v} - \mathbf{x}\|^2 = \|\mathbf{x}\|^2 - \|\mathbf{v}\|^2$ holds, *i.e.,* $\mathbf{v}$ is the tangent point of the tangent line from $\mathbf{x}$ to the surface of $B$.

To make the feasible region of the optimization problem (3) in Theorem 1 nonempty, we need to make an assumption about the positional relationship between $\mathbf{x}$ and the ball $B$. Let $\Pi_H : \mathbb{R}^n \mapsto H$ denote the orthogonal projection from $\mathbb{R}^n$ onto the hyperplane $H$, we make the following assumption:

**Assumption B.1.** $\Pi_H(\mathbf{x}) \in C$

Note that Assumption B.1 is not really an "assumption": it essentially means that there is a tangent point on $S \cap H_{\geq 0}$, which is in the adversarial region. Assumption B.1 means the feasible region of the optimization problem (3) is a nonempty set. By repeatedly reducing the radius $R$, the algorithm guarantees that the optimal tangent point is in the adversarial region, thereby making Assumption B.1 always hold. In addition, according to Assumption B.1, $\|\Pi_H(\mathbf{x})\| \geq R$ holds.

In Theorem 1, $\mathbf{k}$ is an arbitrary point on the surface of the hemisphere $B \cap H_{\geq 0}$, so this proof mainly focuses on points in this region. In the following text, the hemisphere is denoted as $B' := B \cap H_{\geq 0}$, and its surface is denoted as $S' := S \cap H_{\geq 0}$ for brevity. Now, let us pick up any $\mathbf{k} \in S'^2$, and then the intersection point between the hyperplane $H$ and the line passing through $\mathbf{x}$ and $\mathbf{k}$ is denoted as $\mathbf{y_k}$. Then, $(\mathbf{y_k}, \lambda)$ is the unique solution of the following equation system:

$$\begin{aligned} \mathbf{y_k} &= \lambda \mathbf{k} + (1 - \lambda)\mathbf{x}, \\ \langle \mathbf{y_k}, \mathbf{u} \rangle &= 0, \\ 0 &\leq \lambda \leq 1. \end{aligned} \tag{3}$$

Because the position of $\mathbf{k}$ determines the distance between $\mathbf{y_k}$ and $\mathbf{x}$, we can define the function $f(\mathbf{k}) := \|\mathbf{y_k} - \mathbf{x}\|$ to represent the distance between $\mathbf{x}$ and $\mathbf{y_k}$.

## B.2 Proof

To prove Theorem 1, we turn to prove the following lemma, which is equivalent to Theorem 1.

---

[2]Note that $\mathbf{k}$ defined here may not be a tangent point on the ball.

**Lemma 1.** *Let $S'$, $f$ be defined as above, then minimizing $f$ over the feasible region $S'$ is equivalent to finding the point $\mathbf{k}$ from the set $S' \cap C$ that is farthest away from $H$, i.e.,*

$$\arg\min_{\mathbf{k} \in S'} f(\mathbf{k}) = \arg\max_{\mathbf{k} \in (S' \cap C)} \langle \mathbf{k}, \mathbf{u} \rangle. \tag{4}$$

*In addition, we can replace $S'$ with $B'$ in the above equation, and the optimal solution of $f(\mathbf{k})$ does not change. In other words, when the feasible region is $B'$, the optimal solution can be always obtained at the surface of $B'$. Thus, the following equation holds:*

$$\arg\min_{\mathbf{k} \in S'} f(\mathbf{k}) = \arg\min_{\mathbf{k} \in B'} f(\mathbf{k}) = \arg\max_{\mathbf{k} \in (S' \cap C)} \langle \mathbf{k}, \mathbf{u} \rangle = \arg\max_{\mathbf{k} \in (B' \cap C)} \langle \mathbf{k}, \mathbf{u} \rangle. \tag{5}$$

*Proof.* By simplifying the original problem to a two-dimensional plane, the proof of Lemma 1 will be readily apparent. Let $V := \mathrm{span}(\{\mathbf{x}, \mathbf{u}\})$ be the plane spanned by $\mathbf{x}$ and $\mathbf{u}$. It is easy to observe $B$, $S$, $C$, $B'$, and $S'$ are symmetrical about the plane $V$. Next, we will show that for any $\mathbf{k} \in B'$, there must exist a point $\mathbf{k}^* \in S' \cap V$ such that $f(\mathbf{k}^*) \leq f(\mathbf{k})$. To find the $\mathbf{k}^*$ that satisfies the condition, we introduce the notation $\Pi_V : \mathbb{R}^n \mapsto V$ to denote the projection from $\mathbb{R}^n$ to $V$.

Now, take any $\mathbf{k} \in B'$, and use $\mathbf{k}''$ to denote the mirror point of $\mathbf{k}$ with respect to $V$, as shown in Fig. 2. The projection point $\Pi_V(\mathbf{k})$ is the midpoint of the line between $\mathbf{k}$ and $\mathbf{k}''$, *i.e.*, $\mathbf{k}'' = 2\Pi_V(\mathbf{k}) - \mathbf{k}$. Note that if $\mathbf{k} \in V$, then $\mathbf{k}$, $\mathbf{k}'$ and $\mathbf{k}''$ coincide. Since $B'$ is symmetrical about the plane $V$, we have $\mathbf{k}'' \in B'$. Now since $B'$ is the intersection of two convex sets $B$ and $H_{\geq 0}$, we know that $B'$ is also a convex set. Notice that $\Pi_V(\mathbf{k}) = \frac{1}{2} \cdot (\mathbf{k} + \mathbf{k}'')$ is a convex combination of $\mathbf{k}$ and $\mathbf{k}''$, and $B'$ is a convex set, thus we conclude $\Pi_V(\mathbf{k}) \in B'$.

Now, we will show that we can ignore any point outside of $V$, thus restricting the problem to the two-dimensional plane $V$. Formally, the following inequality holds for any $\mathbf{k}$:

$$f(\Pi_V(\mathbf{k})) \leq f(\mathbf{k}). \tag{6}$$

The above inequality is easy to prove. Because $\mathbf{x} \in V$, we have $\|\Pi_V(\mathbf{y_k} - \mathbf{x})\| = \|\Pi_V(\mathbf{y_k}) - \mathbf{x}\|$. Therefore,

$$f(\Pi_V(\mathbf{k})) = \|\Pi_V(\mathbf{y_k}) - \mathbf{x}\| = \|\Pi_V(\mathbf{y_k} - \mathbf{x})\| \leq \|\mathbf{y_k} - \mathbf{x}\| = f(\mathbf{k}). \tag{7}$$

Now, we can focus on the plane $V$ and find the optimal $\mathbf{k}^*$ on it such that $f(\mathbf{k}^*) \leq f(\Pi_V(\mathbf{k}))$. Let us define $C_0$ to denote the convex cone with the point $\mathbf{x}$ as the vertex, and its boundary is formed by all tangent lines from $\mathbf{x}$ to $B$:

$$C_0 := \left\{ \mathbf{v} \in \mathbb{R}^n \mid \cos(\mathbf{v} - \mathbf{x}, -\mathbf{x}) \geq \sqrt{1 - \frac{R^2}{\|\mathbf{x}\|^2}} \right\}. \tag{8}$$

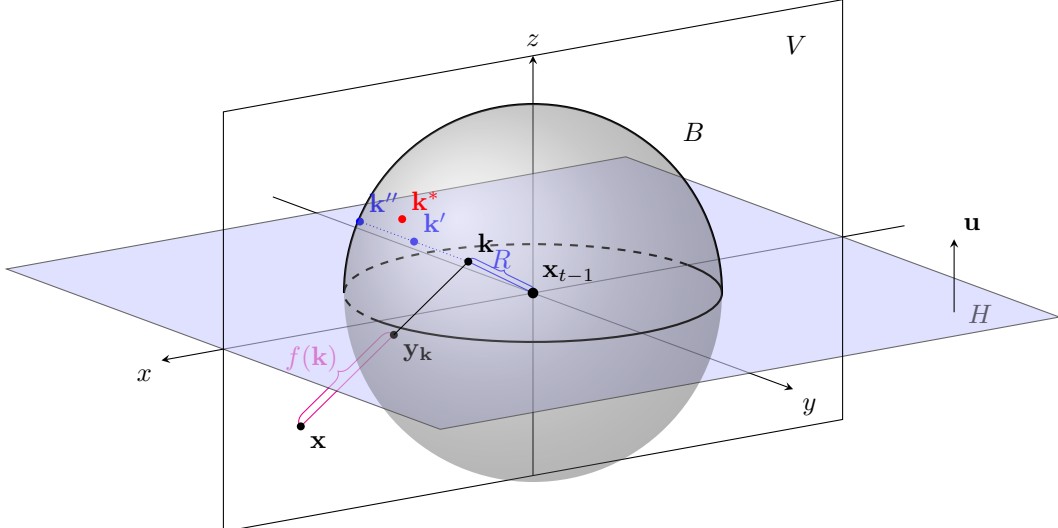

Figure 2: Illustration of the points used in proving Lemma 1, where $\mathbf{k}''$ is the mirror point of $\mathbf{k}$ with respect to the plane $V$, and $\mathbf{k}'$ is the projection of $\mathbf{k}$ onto the plane $V$.

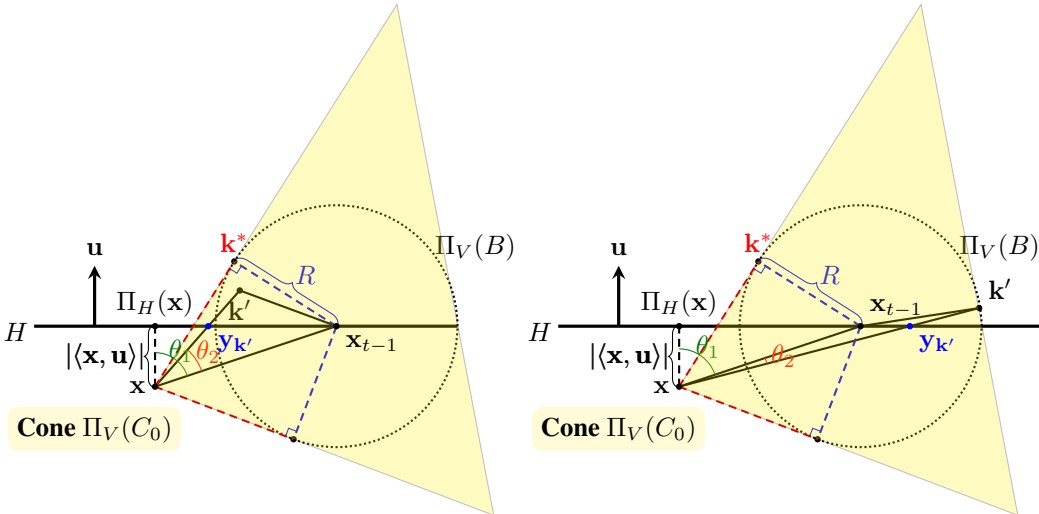

(a) $\mathbf{y_{k'}}$ and $\Pi_H(\mathbf{x})$ are on the same side of $\mathbf{x}_{t-1}$.    (b) $\mathbf{y_{k'}}$ and $\Pi_H(\mathbf{x})$ are on different sides of $\mathbf{x}_{t-1}$.

Figure 3: Illustration of the problem reduced to the plane $V$.

Let $\mathbf{k}' := \Pi_V(\mathbf{k})$ be the projection point of $\mathbf{k}$ onto the plane $V$ (see Fig. 3). Because $\mathbf{k}' \in B'$ and $\mathbf{k}' \in V$, we have $\mathbf{k}' \in \Pi_V(B')$. Now, we define $\mathbf{k}^* \in S' \cap C_0 \cap V$ to be the tangent point from $\mathbf{x}$ to the semicircle $\Pi_V(B')$. We claim $\mathbf{k}^*$ is the optimal one that attains the minimum $f(\mathbf{k}')$ among all $\mathbf{k}'$ in $\Pi_V(B')$. We denote the angle between $-\mathbf{x}$ and $\mathbf{u}$ as $\theta_1$, *i.e.,* $\theta_1 := \arccos(\cos(-\mathbf{x}, \mathbf{u}))$. The angle between $-\mathbf{x}$ and $\mathbf{k}' - \mathbf{x}$ is denoted as $\theta_2$, *i.e.,* $\theta_2 := \arccos(\cos(-\mathbf{x}, \mathbf{k}' - \mathbf{x}))$. Based on the position of $\mathbf{k}'$ in $\Pi_V(B')$, there are two possible cases for the angle $\theta_2$, as shown in Fig. 3a and Fig. 3b, respectively. We discuss them separately below.

In the first case (Fig. 3a), $\mathbf{y_{k'}}$ and $\Pi_H(\mathbf{x})$ are on the same side of $\mathbf{x}_{t-1}$. By Assumption B.1, we know that $\|\Pi_H(\mathbf{x})\| \geq R$, so $\cos(-\mathbf{x}, \mathbf{u}) = \sqrt{1 - \|\Pi_H(\mathbf{x})\|^2 / \|\mathbf{x}\|^2} \leq \sqrt{1 - R^2 / \|\mathbf{x}\|^2}$. According to the definition of the convex cone $C_0$, $\mathbf{u}$ is outside $C_0$. Notice that $\mathbf{x} \in C_0$ and $\mathbf{k}' \in \Pi_V(B')$, hence $\mathbf{k}' - \mathbf{x}$ is in the convex cone $\Pi_V(C_0)$. Therefore, based on the positions of the two vectors $\mathbf{u}$ and $\mathbf{k}' - \mathbf{x}$ with respect to the cone $\Pi_V(C_0)$, we conclude that $\theta_1 \geq \theta_2$. In such case, the distance function is $f(\mathbf{k}') = \|\mathbf{y_{k'}} - \mathbf{x}\| = |\langle \mathbf{x}, \mathbf{u} \rangle| / \cos(\theta_1 - \theta_2)$, as shown in Fig. 3a. Because both $\mathbf{x}$ and $\mathbf{u}$ are fixed, the value of $\theta_1$ is fixed. Therefore, the only way to minimize $f(\mathbf{k}')$ is to maximize $\theta_2$. Among all possible choices of $\mathbf{k}'$ in $\Pi_V(B')$, the $\mathbf{k}'$ that maximizes the angle $\theta_2$ appears on the boundary of $\Pi_V(C_0) \cap H_{\geq 0}$. The only point that satisfies this condition is the tangent point $\mathbf{k}^*$. Therefore, in the first case, $\arg\min_{\mathbf{k} \in B'} f(\mathbf{k}) = \mathbf{k}^*$.

In the second case (Fig. 3b), $\mathbf{y_{k'}}$ and $\Pi_H(\mathbf{x})$ are on different sides of $\mathbf{x}_{t-1}$. In this case, $\theta_2 \geq 0$. In particular, when $\theta_2 = 0$, $\mathbf{y_{k'}}$ and $\mathbf{x}_{t-1}$ coincide. According to Assumption B.1, $\theta_1 > 0$. The distance function can be defined as $f(\mathbf{k}') = \|\mathbf{y_{k'}} - \mathbf{x}\| = |\langle \mathbf{x}, \mathbf{u} \rangle| / \cos(\theta_1 + \theta_2)$ in this case. Because $\theta_1 > 0$ and $\theta_2 \geq 0$, the following inequality holds:

$$f(\mathbf{k}') = \frac{|\langle \mathbf{x}, \mathbf{u} \rangle|}{\cos(\theta_1 + \theta_2)} \geq \frac{|\langle \mathbf{x}, \mathbf{u} \rangle|}{\cos(\theta_1)} \geq \frac{|\langle \mathbf{x}, \mathbf{u} \rangle|}{\cos(\theta_1 - \theta_2)}. \tag{9}$$

According to the above inequality, the distance obtained from the second case is greater than or equal to the distance in the first case, and the distances in both cases are equal only if $\theta_2 = 0$. Therefore, we can still conclude that $f(\mathbf{k}') \geq f(\mathbf{k}^*)$, *i.e.,* $\arg\min_{\mathbf{k} \in B'} f(\mathbf{k}) = \mathbf{k}^*$.

Finally, we need to prove $\arg\max_{\mathbf{k} \in (B' \cap C)} \langle \mathbf{k}, \mathbf{u} \rangle = \mathbf{k}^*$, so that Eq. (5) holds. The overall proof process is similar to the above proof, except that all $f(\mathbf{k})$ in the above proof need to be replaced by $\langle \mathbf{k}, \mathbf{u} \rangle$. Correspondingly, Eq. (6) needs to be changed to the following formula:

$$\langle \mathbf{k}^*, \mathbf{u} \rangle \geq \langle \Pi_V(\mathbf{k}), \mathbf{u} \rangle = \langle \mathbf{k}, \mathbf{u} \rangle. \tag{10}$$

Firstly, let us prove the equality part of Eq. (10): when projecting any $\mathbf{k} \in (B' \cap C)$ onto the plane $V$, the value of $\langle \mathbf{k}, \mathbf{u} \rangle$ does not change. Thus, we have $\langle \Pi_V(\mathbf{k}), \mathbf{u} \rangle = \langle \mathbf{k}, \mathbf{u} \rangle$. Secondly, we prove the inequality part of Eq. (10): $\langle \mathbf{k}^*, \mathbf{u} \rangle \geq \langle \Pi_V(\mathbf{k}), \mathbf{u} \rangle$. Now the problem is reduced to the plane $V$ again.

Because $\Pi_V(\mathbf{k}) \in (B' \cap C \cap V)$, only the first case mentioned above can happen (Fig. 3a). By a similar argument, we conclude that $\arg\max_{\mathbf{k} \in (B' \cap C)} \langle \mathbf{k}, \mathbf{u} \rangle = \mathbf{k}^*$ holds, which proves Lemma 1. Consequently, Theorem 1 holds. $\qquad\square$

## C  Experimental Settings

In this section, we provide the hyperparameter settings of the compared methods, *i.e.,* Hop-SkipJumpAttack (HSJA) [2], Boundary Attack (BA) [1], Sign-OPT [3], and SVM-OPT [3]. In addition, the proposed Tangent Attack is abbreviated as TA, and the Generalized Tangent Attack is abbreviated as G-TA.

Table 1: The hyperparameters of HSJA.

| Dataset | Hyperparameter | Value |
|---|---|---|
| CIFAR-10 | $\gamma$, threshold of the binary search | 1.0 |
| | $B_0$, the initial batch size for gradient estimation | 100 |
| | $B_{\max}$, the maximum batch size for gradient estimation | 10,000 |
| | the search method for step size | geometric progression |
| | number of iterations | 64 |
| ImageNet | $\gamma$, threshold of the binary search | 1,000.0 |
| | $B_0$, the initial batch size for gradient estimation | 100 |
| | $B_{\max}$, the maximum batch size for gradient estimation | 10,000 |
| | the search method for step size | geometric progression |
| | number of iterations | 64 |

Table 2: The hyperparameters of BA.

| Hyperparameter | Value |
|---|---|
| maximum number of trials per iteration | 25 |
| number of iterations | 1,200 |
| spherical step size | 0.01 |
| source step size | 0.01 |
| step size adaptation multiplier | 1.5 |
| disable automatic batch size tuning | False |
| generate candidates and random numbers without using multithreading | False |

Table 3: The hyperparameters of Sign-OPT.

| Hyperparameter | Value |
|---|---|
| $k$, number of queries for estimating an approximate gradient | 200 |
| $\alpha$, the update step size of the direction $\theta$ | 0.2 |
| $\beta$, used for the gradient estimation of $\theta$ and determining the stopping threshold of binary search | 0.001 |
| the number of iterations | 1,000 |
| the binary search's stopping threshold of the CIFAR-10 dataset | $\frac{\beta}{500}$ |
| the binary search's stopping threshold of the ImageNet dataset | $1 \times 10^{-4}$ |

**Experimental Equipment.** The experiments of all compared methods are conducted by using PyTorch 1.7.1 framework on a NVIDIA 1080Ti GPU.

**HSJA.** Hyperparameters of HSJA [2] are listed in Table 1. We translate the implementation code into the PyTorch version for the experiments. In the experiments of targeted attacks, we randomly select an image from the target class as the initial adversarial example. For fair comparison, we set the hyperparameters of TA and G-TA to be the same with HSJA, *i.e.,* the same initial batch size $B_0$ and the same $\gamma$.

**BA.** Hyperparameters of BA [1] are listed in Table 2. In the experiments, we directly use the implementation of BA from Foolbox 2.0 [8, 9], and adopt a randomly selected image from the target class as the initialization in the targeted attack.

Table 4: The hyperparameters of SVM-OPT.

| Hyperparameter | Value |
|---|---|
| $k$, number of queries for estimating gradients | 100 |
| $\alpha$, the step size of the gradient descent of $\theta$ | 0.2 |
| $\beta$, used for the gradient estimation of $\theta$ and determining the stopping threshold of binary search | 0.001 |
| the number of iterations | 1,000 |
| the binary search's stopping threshold of the CIFAR-10 dataset | $\frac{\beta}{500}$ |
| the binary search's stopping threshold of the ImageNet dataset | $1 \times 10^{-4}$ |

**Sign-OPT and SVM-OPT.** Hyperparameters of Sign-OPT [3] and SVM-OPT [3] are listed in Tables 3 and 4. We translate the implementation code into the PyTorch version for the experiments. In the experiments of targeted attacks, we set the initial direction $\theta_0$ of Sign-OPT and SVM-OPT to the direction of a randomly selected image of the target class.

# D   Experimental Results

## D.1   Limitation of Tangent Attack

The proposed approach supports all types of attacks, including both untargeted and targeted attacks under the both $\ell_2$ and $\ell_\infty$ norm constraints. This is the strength of the proposed approach. However, in the $\ell_\infty$ norm attack, TA and G-TA obtain the similar performance to the baseline method HSJA. Because under the definition of the $\ell_\infty$ norm distance: $D_{\ell_\infty}(x, y) := \max_i(|x_i - y_i|), i \in \{1, \ldots, d\}$ ($d$ is the image dimension), the intersection of the tangent line and the decision boundary may not be the one with the shortest $\ell_\infty$ norm distance to the original image. Therefore, searching the boundary sample along the tangent line cannot always outperform HSJA in the $\ell_\infty$ norm attack.

Tables 5 and 6 demonstrate the experimental results of attacking against undefended models on the CIFAR-10 and ImageNet datasets.

Table 5: Mean $\ell_\infty$ distortions of different query budgets on the ImageNet dataset, where the radius ratio $r$ is set to 1.1 in G-TA. BA is not applicable to the $\ell_\infty$ norm attack, hence it is not listed.

| Target Model | Method | Targeted Attack | | | | | | Untargeted Attack | | | | | |
|---|---|---|---|---|---|---|---|---|---|---|---|---|---|
| | | @300 | @1K | @2K | @5K | @8K | @10K | @300 | @1K | @2K | @5K | @8K | @10K |
| Inception-v3 | Sign-OPT [3] | 0.557 | 0.519 | 0.481 | 0.421 | 0.390 | 0.375 | 1.078 | 0.792 | 0.548 | 0.328 | 0.262 | 0.239 |
| | SVM-OPT [3] | 0.558 | 0.512 | 0.476 | 0.423 | 0.397 | 0.385 | 1.079 | 0.763 | 0.526 | 0.336 | 0.280 | 0.260 |
| | HSJA [2] | 0.370 | 0.330 | **0.289** | **0.211** | **0.169** | **0.147** | 0.305 | 0.236 | 0.174 | **0.093** | 0.069 | **0.059** |
| | TA | 0.370 | 0.330 | 0.291 | 0.216 | 0.172 | 0.149 | **0.304** | **0.234** | **0.173** | 0.093 | 0.068 | **0.059** |
| | G-TA | **0.364** | **0.326** | **0.289** | 0.220 | 0.179 | 0.159 | **0.304** | 0.238 | 0.174 | **0.093** | 0.068 | **0.059** |
| Inception-v4 | Sign-OPT [3] | 0.545 | 0.504 | 0.464 | 0.402 | 0.370 | 0.355 | 1.176 | 0.867 | 0.603 | 0.369 | 0.296 | 0.270 |
| | SVM-OPT [3] | 0.547 | 0.498 | 0.460 | 0.406 | 0.379 | 0.367 | 1.181 | 0.842 | 0.588 | 0.381 | 0.319 | 0.296 |
| | HSJA [2] | 0.357 | **0.324** | **0.287** | **0.215** | **0.175** | **0.152** | **0.336** | 0.257 | 0.185 | 0.091 | 0.060 | **0.048** |
| | TA | **0.354** | 0.328 | 0.294 | 0.221 | 0.182 | 0.161 | 0.337 | 0.264 | 0.196 | 0.103 | 0.073 | 0.062 |
| | G-TA | **0.354** | **0.324** | 0.290 | 0.220 | 0.182 | 0.162 | 0.337 | 0.264 | 0.196 | 0.104 | 0.073 | 0.061 |
| SENet-154 | Sign-OPT [3] | 0.537 | 0.491 | 0.439 | 0.357 | 0.316 | 0.298 | 0.806 | 0.631 | 0.462 | 0.299 | 0.247 | 0.227 |
| | SVM-OPT [3] | 0.538 | 0.480 | 0.429 | 0.357 | 0.322 | 0.307 | 0.807 | 0.608 | 0.444 | 0.306 | 0.262 | 0.246 |
| | HSJA [2] | 0.347 | **0.288** | **0.249** | **0.176** | **0.139** | **0.119** | **0.253** | **0.195** | **0.141** | 0.071 | 0.048 | 0.039 |
| | TA | 0.346 | 0.289 | 0.251 | 0.181 | 0.142 | 0.123 | **0.253** | 0.196 | **0.141** | 0.071 | **0.047** | **0.038** |
| | G-TA | **0.344** | **0.288** | 0.252 | 0.179 | 0.142 | 0.123 | **0.253** | 0.196 | **0.141** | 0.070 | **0.047** | **0.038** |
| ResNet-101 | Sign-OPT [3] | 0.549 | 0.501 | 0.450 | 0.370 | 0.329 | 0.309 | 0.645 | 0.515 | 0.385 | 0.251 | 0.206 | 0.190 |
| | SVM-OPT [3] | 0.550 | 0.492 | 0.444 | 0.371 | 0.335 | 0.319 | 0.642 | 0.494 | 0.370 | 0.258 | 0.220 | 0.206 |
| | HSJA [2] | 0.340 | 0.283 | 0.247 | **0.179** | **0.143** | **0.125** | 0.197 | **0.140** | 0.098 | 0.049 | 0.034 | 0.028 |
| | TA | 0.340 | 0.282 | **0.246** | 0.180 | **0.143** | **0.125** | **0.196** | 0.145 | 0.109 | 0.064 | 0.050 | 0.044 |
| | G-TA | **0.337** | **0.280** | **0.246** | 0.182 | 0.150 | 0.132 | **0.196** | 0.147 | 0.110 | 0.064 | 0.050 | 0.044 |

The results of Tables 5 and 6 show that HSJA, TA and G-TA obtain the similar average $\ell_\infty$ distortions. Therefore, although the proposed approach is applicable to $\ell_\infty$ norm attack, the performance is similar to that of the baseline method HSJA.

Table 6: Mean $\ell_\infty$ distortions of different query budgets on the CIFAR-10 dataset, where the radius ratio $r$ is set to 1.5 in G-TA. BA is not applicable to the $\ell_\infty$ norm attack, and thus it is not listed.

| Target Model | Method | Targeted Attack | | | | | | Untargeted Attack | | | | | |
|---|---|---|---|---|---|---|---|---|---|---|---|---|---|
| | | @300 | @1K | @2K | @5K | @8K | @10K | @300 | @1K | @2K | @5K | @8K | @10K |
| PyramidNet-272 | Sign-OPT [3] | 0.395 | 0.318 | 0.237 | 0.134 | 0.096 | 0.082 | 0.284 | 0.189 | 0.115 | 0.059 | 0.047 | 0.043 |
| | SVM-OPT [3] | 0.390 | 0.299 | 0.226 | 0.134 | 0.099 | 0.087 | 0.286 | 0.173 | 0.104 | 0.059 | 0.049 | 0.046 |
| | HSJA [2] | **0.218** | 0.155 | **0.112** | **0.057** | **0.039** | 0.032 | **0.133** | **0.056** | **0.034** | **0.016** | **0.012** | 0.011 |
| | TA | 0.219 | 0.154 | **0.112** | **0.057** | **0.039** | 0.032 | 0.134 | 0.057 | 0.035 | 0.017 | 0.013 | 0.011 |
| | G-TA | **0.218** | **0.153** | 0.113 | **0.057** | **0.039** | **0.031** | **0.133** | **0.056** | **0.034** | **0.016** | **0.012** | **0.010** |
| GDAS | Sign-OPT [3] | 0.398 | 0.332 | 0.266 | 0.153 | 0.107 | 0.089 | 0.305 | 0.269 | 0.231 | 0.185 | 0.167 | 0.160 |
| | SVM-OPT [3] | 0.389 | 0.325 | 0.267 | 0.164 | 0.118 | 0.100 | 0.304 | 0.257 | 0.219 | 0.181 | 0.168 | 0.163 |
| | HSJA [2] | **0.210** | **0.147** | **0.112** | **0.060** | **0.040** | 0.031 | **0.049** | **0.029** | **0.020** | **0.011** | **0.009** | **0.008** |
| | TA | 0.214 | 0.151 | 0.115 | 0.062 | 0.041 | 0.032 | **0.049** | **0.029** | **0.020** | **0.011** | **0.009** | **0.008** |
| | G-TA | 0.214 | 0.151 | 0.116 | 0.062 | 0.041 | 0.032 | **0.049** | **0.029** | **0.020** | **0.011** | **0.009** | **0.008** |
| WRN-28 | Sign-OPT [3] | 0.402 | 0.307 | 0.225 | 0.121 | 0.086 | 0.074 | 0.200 | 0.130 | 0.085 | 0.053 | 0.044 | 0.041 |
| | SVM-OPT [3] | 0.382 | 0.296 | 0.223 | 0.128 | 0.093 | 0.080 | 0.201 | 0.121 | 0.079 | 0.052 | 0.045 | 0.043 |
| | HSJA [2] | **0.185** | **0.106** | 0.070 | 0.032 | **0.021** | 0.018 | **0.090** | 0.031 | **0.020** | **0.012** | **0.010** | **0.009** |
| | TA | 0.186 | 0.107 | 0.070 | **0.031** | 0.021 | **0.018** | **0.090** | 0.030 | **0.020** | **0.012** | **0.010** | **0.009** |
| | G-TA | **0.185** | **0.106** | **0.069** | 0.032 | 0.022 | **0.018** | **0.090** | **0.030** | **0.020** | **0.012** | **0.010** | **0.009** |
| WRN-40 | Sign-OPT [3] | 0.397 | 0.305 | 0.220 | 0.120 | 0.085 | 0.073 | 0.284 | 0.208 | 0.125 | 0.051 | 0.042 | 0.039 |
| | SVM-OPT [3] | 0.381 | 0.293 | 0.220 | 0.126 | 0.092 | 0.079 | 0.273 | 0.190 | 0.120 | 0.057 | 0.045 | 0.041 |
| | HSJA [2] | **0.194** | 0.111 | **0.072** | 0.032 | 0.022 | 0.019 | 0.084 | 0.030 | **0.020** | **0.012** | **0.010** | **0.009** |
| | TA | 0.195 | 0.112 | 0.073 | 0.032 | 0.022 | 0.019 | **0.082** | **0.029** | **0.020** | **0.012** | **0.010** | **0.009** |
| | G-TA | **0.194** | 0.110 | **0.072** | 0.032 | 0.022 | 0.019 | **0.082** | **0.029** | **0.020** | **0.012** | **0.010** | **0.009** |

## D.2 Experimental Results of Attacks against Defense Models

We also conduct experiments by using $\ell_\infty$ norm attacks to break five defense models, and the experimental results are shown in Table 7. The conclusion drawn from this table is the same as that in Tables 5 and 6: TA and G-TA obtain the similar performance with HSJA in $\ell_\infty$ norm attacks.

Table 7: The experimental results of performing $\ell_\infty$ norm attacks against the defense models on the CIFAR-10 dataset, where the radius ratio $r$ is set to 1.5 in G-TA.

| Target Model | Method | Untargeted Attack | | | | | |
|---|---|---|---|---|---|---|---|
| | | @300 | @1K | @2K | @5K | @8K | @10K |
| AT [7] | Sign-OPT [3] | 0.731 | 0.519 | 0.395 | 0.288 | 0.255 | 0.243 |
| | SVM-OPT [3] | 0.719 | 0.498 | 0.382 | 0.287 | 0.261 | 0.251 |
| | HSJA [2] | **0.181** | **0.145** | **0.121** | **0.090** | 0.080 | **0.075** |
| | TA | 0.184 | 0.147 | **0.121** | **0.090** | 0.079 | **0.075** |
| | G-TA | **0.181** | **0.145** | **0.121** | **0.090** | 0.080 | **0.075** |
| TRADES [11] | Sign-OPT [3] | 0.748 | 0.562 | 0.419 | 0.304 | 0.269 | 0.257 |
| | SVM-OPT [3] | 0.743 | 0.534 | 0.409 | 0.308 | 0.281 | 0.271 |
| | HSJA [2] | **0.194** | **0.162** | **0.137** | **0.106** | **0.095** | **0.090** |
| | TA | 0.195 | 0.163 | 0.138 | 0.107 | **0.095** | **0.090** |
| | G-TA | **0.194** | 0.163 | 0.138 | 0.107 | **0.095** | **0.090** |
| JPEG [4] | Sign-OPT [3] | 0.301 | 0.292 | 0.281 | 0.262 | 0.250 | 0.245 |
| | SVM-OPT [3] | 0.301 | 0.288 | 0.275 | 0.256 | 0.249 | 0.246 |
| | HSJA [2] | 0.094 | **0.086** | **0.078** | **0.066** | **0.061** | **0.058** |
| | TA | **0.093** | 0.087 | 0.080 | 0.067 | **0.061** | **0.058** |
| | G-TA | 0.097 | 0.091 | 0.081 | 0.068 | 0.062 | 0.059 |
| Feature Distillation [6] | Sign-OPT [3] | 0.344 | 0.330 | 0.317 | 0.290 | 0.273 | 0.266 |
| | SVM-OPT [3] | 0.354 | 0.338 | 0.323 | 0.297 | 0.284 | 0.279 |
| | HSJA [2] | 0.090 | 0.087 | 0.080 | 0.069 | 0.064 | 0.061 |
| | TA | **0.089** | **0.086** | **0.079** | 0.070 | 0.063 | 0.060 |
| | G-TA | 0.090 | **0.086** | **0.079** | **0.067** | **0.062** | **0.059** |
| Feature Scatter [10] | Sign-OPT [3] | 0.561 | 0.380 | 0.246 | 0.135 | 0.110 | 0.101 |
| | SVM-OPT [3] | 0.550 | 0.344 | 0.222 | 0.137 | 0.116 | 0.110 |
| | HSJA [2] | **0.202** | **0.137** | **0.104** | **0.062** | **0.048** | **0.042** |
| | TA | **0.202** | **0.137** | **0.104** | **0.062** | **0.048** | **0.042** |
| | G-TA | 0.205 | 0.139 | 0.105 | **0.062** | **0.048** | **0.042** |

Next, we conduct experiments by using $\ell_2$ norm attack to break different defense models on the CIFAR-10 and ImageNet datasets. In the CIFAR-10 dataset, we select six types of defense models:

- Adversarial Training (AT) [7]: the most effective defense method, which uses adversarial examples as the training data to obtain the robust classifier.

- TRADES [11]: an improved AT that optimizes a regularized surrogate loss.

- JPEG [4]: a standard image compression algorithm based on the discrete cosine transform, which can remove the adversarial perturbations, thereby providing some degree of defense.

- Feature Distillation [6]: a defense method based on the improved JPEG image compression. Its defense mechanism is divided into two steps. Firstly, it filters out adversarial perturbations by using a semi-analytical method. Secondly, it restores the classification accuracy of benign images by using a DNN-oriented quantization process.

- Feature Scatter [10]: a feature scattering-based AT method, which is an unsupervised approach for generating adversarial examples during the training.

- ComDefend [5]: a defense model that consists of a compression CNN and a reconstruction CNN to transform the adversarial image into its clean version to defend against attacks.

In the ImageNet dataset, we directly use the publicly available AT models for experiments, all of which use the ResNet-50 networks as their backbones. The pre-trained weights can be downloaded from `https://github.com/MadryLab/robustness`. In the experiments, we set the radius ratio $r$ of G-TA to 1.5, and the experimental results are shown in Fig. 4. In untargeted attacks (Figs. 4a, 4b, 4c), the G-TA (the semi-ellipsoid version) outperforms the TA (the hemisphere version), and the baseline method HSJA outperforms TA and G-TA. We conjecture that it is because the classification decision boundaries of the AT models on the ImageNet dataset are extremely curved in untargeted attacks, resulting in the better performance of HSJA. In targeted attacks (Figs. 4d, 4e, 4f), both TA and G-TA outperform HSJA in the attacks of different AT models. These results indicate that TA and G-TA are more suitable for the targeted attack. Another interesting finding is that SVM-OPT performs better in untargeted attacks while Sign-OPT performs better in targeted attacks. We will explore the reasons for these results in the future work.

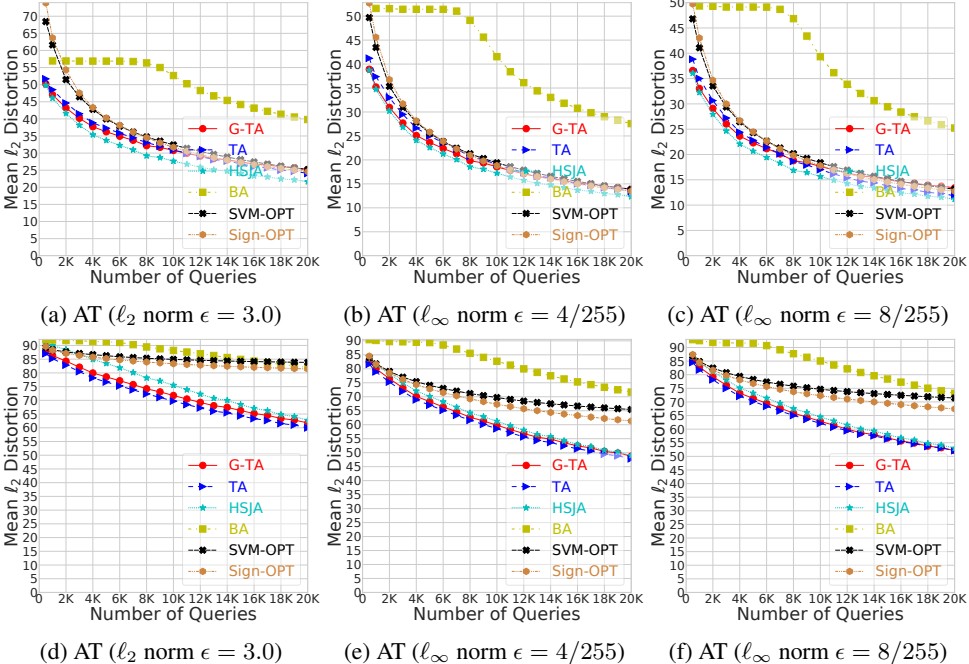

(a) AT ($\ell_2$ norm $\epsilon = 3.0$)  (b) AT ($\ell_\infty$ norm $\epsilon = 4/255$)  (c) AT ($\ell_\infty$ norm $\epsilon = 8/255$)

(d) AT ($\ell_2$ norm $\epsilon = 3.0$)  (e) AT ($\ell_\infty$ norm $\epsilon = 4/255$)  (f) AT ($\ell_\infty$ norm $\epsilon = 8/255$)

Figure 4: Experimental results of $\ell_2$ norm attacks against adversarial trained ResNet-50 networks on the ImageNet dataset, where the first row (Figs. 4a, 4b, 4c) shows the results of untargeted attacks, and the second row (Figs. 4d, 4e, 4f) shows the results of targeted attacks.

Figs. 5 and 6 show the experimental results of untargeted and targeted attacks on the CIFAR-10 dataset, respectively. In the results of untargeted attacks (Fig. 5), G-TA outperforms HSJA and TA in the attacks of ComDefend, JPEG and Feature Distillation. When the target models are AT, Feature Scatter and TRADES, the performance of G-TA is similar to that of the baseline attack method HSJA.

In addition, in the experimental results of targeted attacks (Fig. 6), the performance of G-TA is similar to that of TA when attacking different defense models.

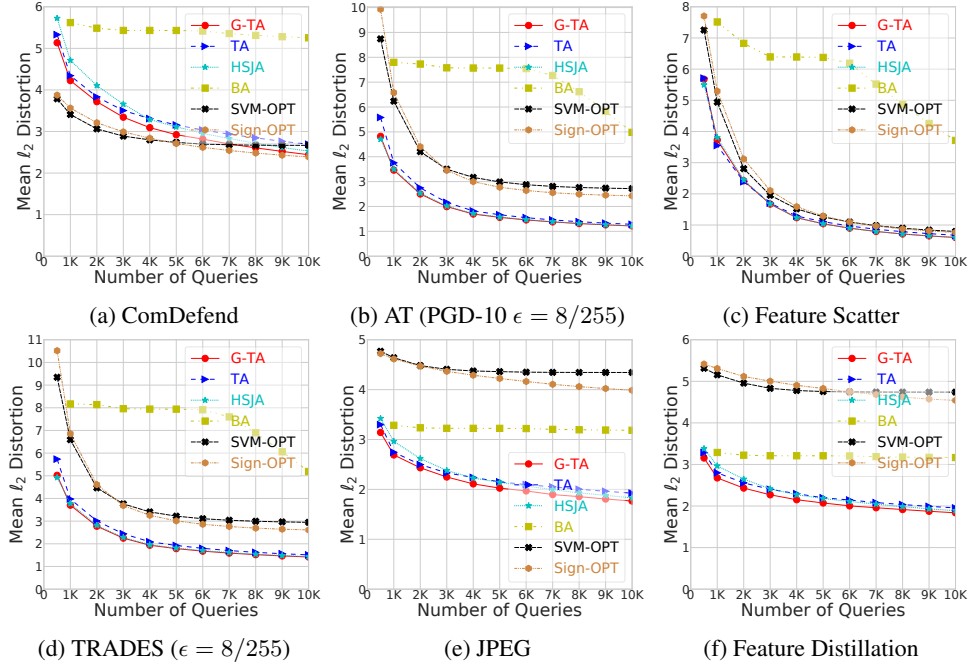

Figure 5: Experimental results of the $\ell_2$ norm untargeted attacks against defense models on the CIFAR-10 dataset, where all defense models adopt the backbone of ResNet-50 network.

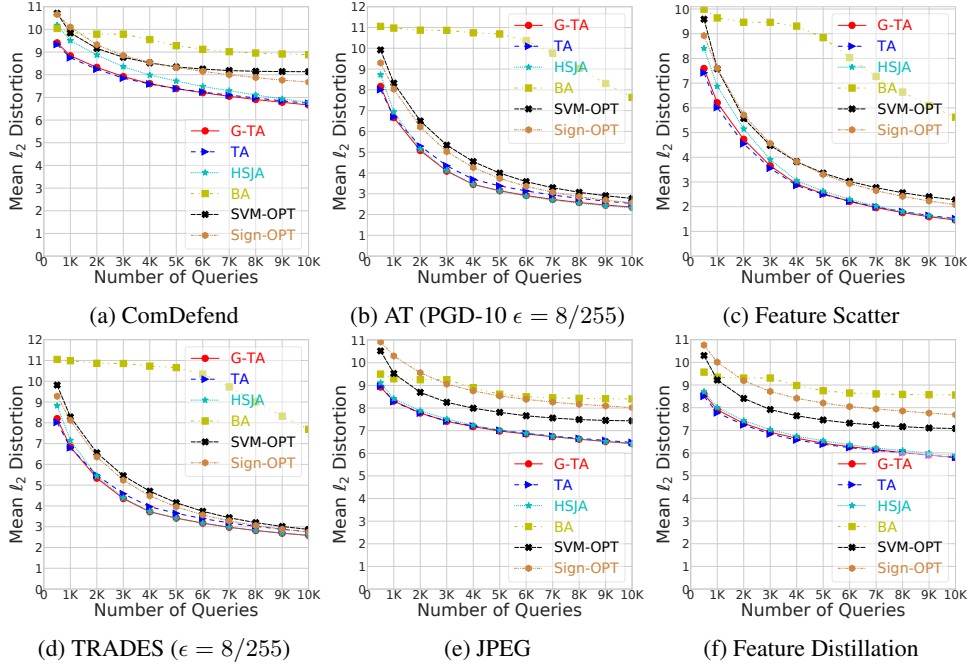

Figure 6: Experimental results of the $\ell_2$ norm targeted attacks against defense models on the CIFAR-10 dataset, where all defense models adopt the backbone of ResNet-50 network.

### D.3 Distributions of Distortions across Different Adversarial Examples

So far, all the experimental results only show the average $\ell_2$ distortion of 1,000 adversarial examples. To check the distortion of each adversarial example in more detail, we extract the $\ell_2$ distortions of 20 samples from HSJA, TA and G-TA. These samples are selected from 1,000 images in the following way: from the 1st image to the 1,000th image, we select one image for every 50 images. Fig. 7 shows the distributions of $\ell_2$ distortions across 20 adversarial examples on the ImageNet dataset, where the

1st image's "image number index" is 0. The results indicate that the $\ell_2$ distortions obtained by TA and G-TA are uniformly better than that of the baseline method HSJA. Thus, our approach can obtain better $\ell_2$ distortions on different adversarial examples, not just on specific samples.

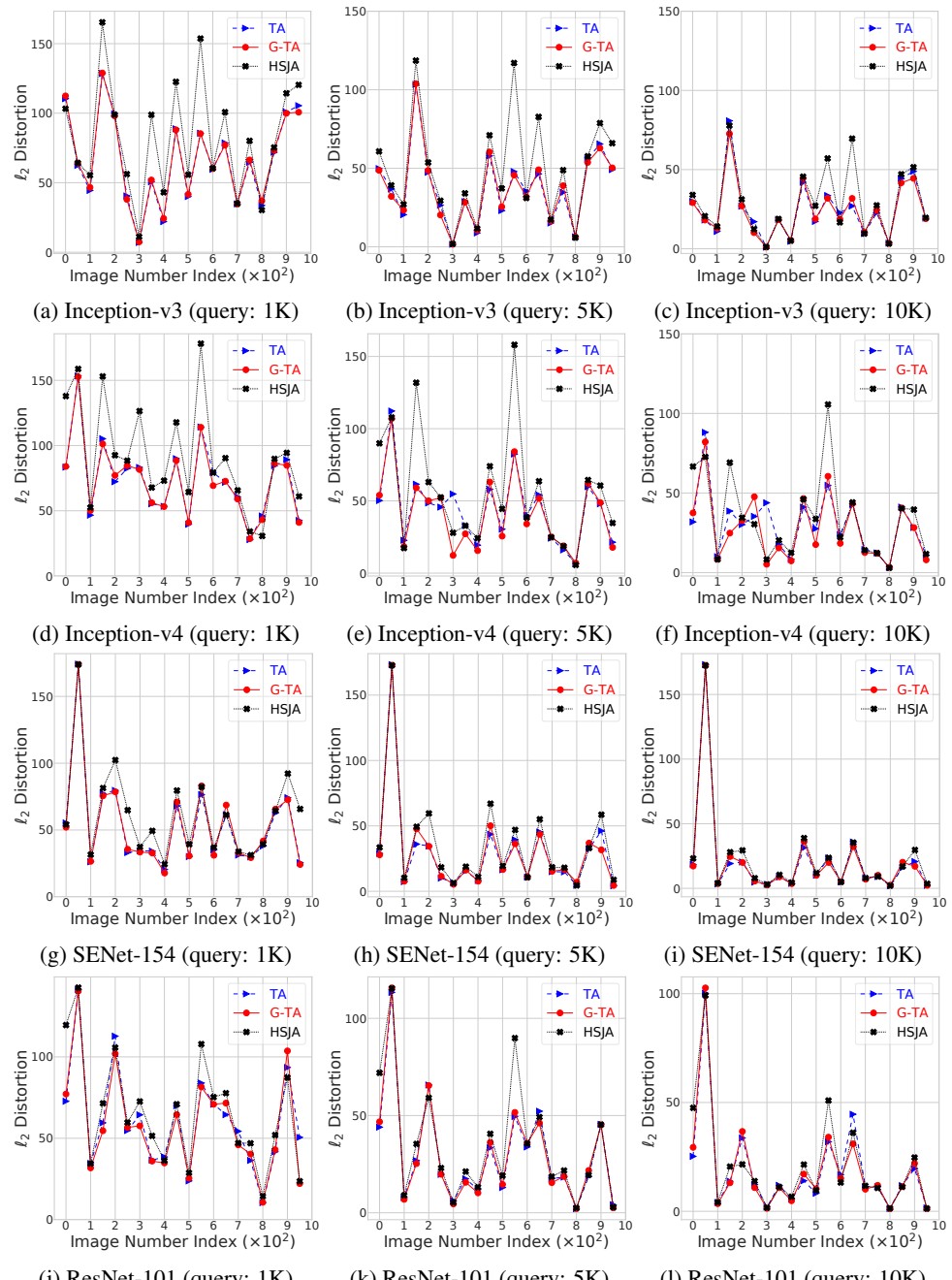

Figure 7: Comparisons of $\ell_2$ distortions across 20 adversarial examples in targeted attacks of the ImageNet dataset.

## D.4 Experimental Results of Median Distortions

In this section, we report the median $\ell_2$ distortions of different query budgets on the CIFAR-10 and ImageNet datasets. Tables 8 and 9 show the experimental results. We can draw the following conclusions based on the results.

Table 8: Median $\ell_2$ distortions of different query budgets on the ImageNet dataset. "-" denotes no adversarial example is found in this query budget.

| Target Model | Method | Targeted Attack | | | | | | Untargeted Attack | | | | | |
|---|---|---|---|---|---|---|---|---|---|---|---|---|---|
| | | @300 | @1K | @2K | @5K | @8K | @10K | @300 | @1K | @2K | @5K | @8K | @10K |
| Inception-v3 | BA [1] | 105.513 | 101.877 | 101.056 | 97.481 | 81.269 | 73.524 | - | 109.507 | 103.637 | 96.340 | 79.027 | 58.924 |
| | Sign-OPT [3] | 96.905 | 83.215 | 66.601 | 43.350 | 31.036 | 25.380 | 115.140 | 73.319 | 38.327 | 12.761 | 8.277 | 6.808 |
| | SVM-OPT [3] | **93.649** | 77.838 | 63.631 | 42.897 | 31.838 | 26.322 | 114.879 | 59.343 | **30.627** | 12.085 | 8.352 | 7.025 |
| | HSJA [2] | 106.341 | 92.114 | 79.225 | 47.469 | 30.624 | 23.838 | 105.702 | 53.880 | 32.684 | 12.360 | 7.829 | 6.227 |
| | TA | 96.612 | 75.610 | 62.573 | **38.226** | **25.892** | **19.993** | **95.302** | **50.878** | 31.833 | 11.921 | **7.464** | **6.030** |
| | G-TA | 97.449 | **75.499** | **62.484** | 38.886 | 26.004 | 20.091 | 96.410 | 50.985 | 31.176 | **11.861** | 7.549 | 6.087 |
| Inception-v4 | BA [1] | 104.275 | 101.115 | 99.872 | 96.700 | 79.412 | 71.387 | - | 116.855 | 112.335 | 104.557 | 85.044 | 64.123 |
| | Sign-OPT [3] | 95.388 | 81.865 | 66.159 | 42.871 | 31.241 | 25.798 | 121.725 | 77.838 | 40.465 | 14.268 | 8.924 | 7.153 |
| | SVM-OPT [3] | **92.640** | 77.616 | 62.949 | 42.552 | 31.142 | 26.238 | 120.407 | 64.600 | **33.960** | **13.586** | 9.035 | 7.535 |
| | HSJA [2] | 104.969 | 90.371 | 78.103 | 47.340 | 31.404 | 24.270 | 109.422 | 60.356 | 37.302 | 14.191 | 8.790 | 6.934 |
| | TA | 96.808 | **74.829** | **61.974** | **37.155** | **26.128** | 21.184 | **101.170** | **55.876** | 36.403 | 14.176 | **8.592** | **6.814** |
| | G-TA | 95.563 | 75.889 | 62.404 | 38.457 | 26.495 | **21.069** | 101.186 | 57.672 | 36.743 | 13.999 | 8.694 | 6.856 |
| SENet-154 | BA [1] | 75.653 | 72.327 | 71.420 | 68.293 | 52.332 | 44.391 | - | 75.355 | 70.498 | 65.186 | 51.950 | 38.164 |
| | Sign-OPT [3] | 70.500 | 59.556 | 45.566 | 27.062 | 18.218 | 14.400 | 65.524 | 42.690 | 23.688 | 9.054 | 5.331 | 4.194 |
| | SVM-OPT [3] | 73.344 | 55.891 | 44.195 | 27.826 | 19.544 | 15.883 | 65.957 | 35.596 | **20.549** | 8.760 | 5.368 | 4.332 |
| | HSJA [2] | 72.589 | 60.361 | 49.487 | 25.718 | 14.929 | 12.197 | 70.043 | 34.697 | 21.811 | 8.098 | 4.482 | 3.707 |
| | TA | 66.285 | **51.012** | **40.475** | **21.590** | **13.293** | **10.782** | **64.784** | 34.034 | 22.269 | **7.636** | 4.273 | 3.555 |
| | G-TA | **66.077** | 51.852 | 41.065 | 21.946 | 13.461 | 10.899 | 65.122 | **33.841** | 21.823 | 7.772 | **4.231** | **3.489** |
| ResNet-101 | BA [1] | 76.772 | 72.674 | 71.761 | 68.231 | 54.847 | 47.785 | - | 63.568 | 59.384 | 55.402 | 42.777 | 29.097 |
| | Sign-OPT [3] | 72.361 | 62.383 | 48.664 | 30.089 | 20.752 | 16.478 | 53.757 | 35.070 | 19.035 | 8.442 | 5.929 | 4.999 |
| | SVM-OPT [3] | 73.758 | 58.716 | 47.496 | 30.443 | 21.502 | 17.535 | 52.471 | 29.225 | 16.469 | 8.245 | 6.043 | 5.259 |
| | HSJA [2] | 73.422 | 60.175 | 49.443 | 26.504 | 16.035 | 12.661 | 54.869 | 24.971 | 15.161 | 6.084 | 3.787 | 3.237 |
| | TA | 69.511 | **55.389** | 44.343 | 24.500 | **14.778** | **11.802** | **51.829** | 24.748 | 15.162 | 5.941 | **3.698** | 3.203 |
| | G-TA | **69.117** | 56.275 | **44.315** | **24.316** | 15.133 | 11.946 | 51.883 | **24.403** | **14.643** | **5.842** | 3.703 | **3.191** |

Table 9: Median $\ell_2$ distortions of different query budgets on the CIFAR-10 dataset.

| Target Model | Method | Targeted Attack | | | | | | Untargeted Attack | | | | | |
|---|---|---|---|---|---|---|---|---|---|---|---|---|---|
| - | | 300 | @1K | @2K | @5K | @8K | @10K | 300 | @1K | @2K | @5K | @8K | @10K |
| PyramidNet-272 | BA [1] | 8.240 | 7.711 | 7.697 | 6.013 | 3.938 | 3.068 | - | 5.133 | 4.268 | 4.060 | 2.471 | 1.460 |
| | Sign-OPT [3] | 7.900 | 6.050 | 3.796 | 1.441 | 0.762 | 0.549 | 3.821 | 1.952 | 0.980 | 0.345 | 0.232 | **0.196** |
| | SVM-OPT [3] | 8.870 | 6.432 | 4.199 | 1.651 | 0.894 | 0.655 | 3.777 | 1.956 | 0.877 | 0.363 | 0.235 | 0.202 |
| | HSJA [2] | 7.616 | 4.013 | 2.109 | **0.589** | 0.384 | 0.325 | 3.935 | **1.022** | **0.587** | 0.294 | 0.224 | 0.201 |
| | TA | 7.650 | **3.874** | **2.071** | 0.599 | **0.380** | 0.318 | **3.758** | 1.028 | 0.589 | 0.289 | **0.223** | 0.197 |
| | G-TA | **7.452** | 3.980 | 2.110 | 0.602 | 0.387 | 0.324 | 3.938 | 1.033 | 0.590 | **0.288** | 0.224 | 0.198 |
| GDAS | BA [1] | 8.098 | 7.568 | 7.554 | 5.774 | 3.301 | 2.396 | - | 2.626 | 2.409 | 2.286 | 1.541 | 1.015 |
| | Sign-OPT [3] | 7.947 | 6.418 | 4.166 | 1.514 | 0.669 | 0.457 | 2.067 | 1.331 | 0.766 | 0.298 | 0.209 | 0.176 |
| | SVM-OPT [3] | 9.138 | 7.242 | 5.090 | 2.103 | 1.043 | 0.673 | 2.043 | 1.230 | 0.674 | 0.302 | 0.211 | 0.183 |
| | HSJA [2] | 7.687 | 3.061 | 1.383 | 0.435 | 0.298 | 0.254 | 1.905 | 0.674 | 0.429 | 0.232 | 0.185 | 0.168 |
| | TA | 7.667 | **3.024** | **1.380** | 0.435 | **0.296** | 0.253 | 1.932 | 0.690 | 0.425 | 0.228 | 0.185 | 0.169 |
| | G-TA | 7.728 | 3.104 | 1.385 | **0.430** | 0.298 | 0.253 | **1.883** | **0.665** | 0.428 | **0.226** | **0.182** | **0.167** |
| WRN-28 | BA [1] | 8.317 | 7.789 | 7.764 | 5.493 | 2.199 | 1.293 | - | 3.900 | 3.332 | 3.167 | 1.361 | 0.732 |
| | Sign-OPT [3] | 7.737 | 5.188 | 2.816 | 0.797 | 0.439 | 0.354 | 2.679 | 1.298 | 0.723 | 0.281 | **0.214** | **0.191** |
| | SVM-OPT [3] | 9.054 | 5.697 | 3.317 | 0.981 | 0.511 | 0.398 | 2.627 | 1.279 | 0.627 | 0.288 | 0.218 | 0.198 |
| | HSJA [2] | 6.446 | 2.064 | 1.005 | 0.443 | 0.339 | **0.306** | 2.497 | 0.697 | 0.442 | 0.264 | 0.224 | 0.208 |
| | TA | 6.518 | **2.018** | **0.988** | 0.428 | 0.337 | **0.306** | 2.606 | **0.682** | **0.431** | 0.262 | 0.225 | 0.210 |
| | G-TA | **6.444** | 2.060 | 1.000 | 0.439 | 0.341 | **0.306** | 2.538 | **0.682** | 0.444 | **0.261** | 0.223 | 0.209 |
| WRN-40 | BA [1] | 8.181 | 7.760 | 7.722 | 5.482 | 2.193 | 1.363 | - | 3.773 | 3.187 | 3.045 | 1.321 | 0.726 |
| | Sign-OPT [3] | 7.782 | 5.285 | 2.895 | 0.845 | 0.455 | 0.369 | 2.510 | 1.213 | 0.679 | 0.265 | **0.194** | **0.173** |
| | SVM-OPT [3] | 9.042 | 5.835 | 3.400 | 1.030 | 0.549 | 0.420 | 2.500 | 1.251 | 0.611 | 0.272 | 0.198 | 0.179 |
| | HSJA [2] | 6.578 | 2.183 | 1.040 | 0.439 | 0.338 | **0.305** | 2.470 | 0.702 | 0.453 | 0.256 | 0.214 | 0.198 |
| | TA | 6.747 | 2.100 | **0.983** | 0.435 | 0.337 | **0.305** | 2.584 | **0.680** | 0.434 | 0.254 | 0.215 | 0.201 |
| | G-TA | **6.514** | **2.069** | 1.014 | 0.438 | 0.339 | 0.306 | **2.453** | 0.695 | 0.441 | 0.255 | 0.213 | 0.199 |

(1) TA and G-TA perform better in attacking high-resolution images, *i.e.,* the images of the ImageNet dataset. The median $\ell_2$ distortions of Table 8 are larger than that of Table 9, because the high-resolution images of the ImageNet dataset lead to larger $\ell_2$ distortions.

(2) TA is more effective in the targeted attacks. We speculate that it is because the adversarial region of the target class is narrower and more scattered in the targeted attack, resulting in a smoother decision boundary. Thus, TA is more suitable for targeted attacks.