# OpenReview forum: "Finding Optimal Tangent Points for Reducing Distortions of Hard-label Attacks"
_NeurIPS.cc/2021/Conference — NeurIPS 2021 Poster_

### Official Review · Reviewer_ZX4s · 2021-07-13

**Rating:** 6
**Confidence:** 4

**Summary:**

This paper discovers that searching the optimal tangent point of a virtual hemisphere could help find minimal adversarial distortion; therefore, this paper proposes an effective method of hard-label attack

**Main Review:**

Pros.

1 The paper is well-written and well-organized. In particular, I like the related work part, which is well-organized.

2 The experiments on standard models and defensive models exhibit the efficacy of the proposal.

Comments.

I have a question about the assumption of the low-curvature smooth surface.
It is well known that traditional adversarial training techniques such as AT and TRADES have the undesirable issue of robust overfitting.
When robust overfitting occurs, the decision boundary becomes more and more curved [1]. Would the overfitted model severely violate the assumption that will weaken the efficacy of the proposal?
Authors may try the friendly adversarial training [2], where it used curriculum training techniques to elude the issue of robust overfitting.

[1] Geometry-aware instance-reweighed adversarial training, in ICLR 2021.

[2] Attacks which do not kill training make adversarial learning stronger, in ICML 2020.

######################## post rebuttal ######################
I have read the author(s)' response and other reviewers' comments. I maintain my scores. Thanks.

**Time Spent Reviewing:**

5 hours

---

> ### Author Response · Authors · 2021-08-10
> **Response to Reviewer R5**
>
> We would like to thank the reviewer for the positive comments, and we shall carefully revise the paper following your suggestions.
>
>
> **Q1**: When robust overfitting occurs, the decision boundary becomes more and more curved. Would the overfitted model severely violate the assumption that will weaken the efficacy of the proposal?
>
> **A1**: The robust overfitting of defended models does not affect the performance of the proposed method,  because neither the derivation nor the algorithm requires flat decision boundary assumption to work. As shown in the derivation of our algorithm, we only need to focus on the decision boundary restricted to a two-dimensional plane (e.g., Figure 1).
> If the decision boundary is curved "downwardly" (as opposed to the example in Fig. 3b), then searching along the tangent line of the hemisphere is still better than the HSJA solution.
> The worst case is when the boundary is curved "upwardly" and has a large curvature (Fig. 3b), which is handled by the semi-ellipsoid variant of our algorithm.
> We replace the hemisphere with a semi-ellipsoid and use a parameter $r$ to control its shape: the semi-ellipsoid becomes "slender" by setting a large $r$, which increases the height of the tangent point $\mathbf{k}$ to deal with this case.
>
> The stable improvement of performance in experimental results of AT and TRADES (Figure 4 in the main text of our paper and Figures 4,5,6 in the Appendix) also proves the effectiveness of this method.

---

### Official Review · Reviewer_c1JB · 2021-07-14

**Rating:** 5
**Confidence:** 4

**Summary:**

In this paper, the authors proposed a new geometric-based method for query-efficient hard-label black-box attacks. The proposed method relies on the observation that the minimum l2 distortion can be obtained by searching a boundary point along a tangent line of a virtual hemisphere. The authors offered a closed-form solution for computing the optimal tangent point and provided a formal proof of its correctness. Lastly, the authors evaluated the proposed approach through extensive experiments.

**Limitations And Societal Impact:**

see above, societal impact not applicable

**Main Review:**

1. This paper proposed a geometric-based method to compute the optimal perturbation direction. However, it seems that the authors assume the decision boundary is a hyperplane H and the gradient direction u is known. This raises my two concerns. First, H can be seen as a hyperplane only when we are considering a very small local region. Yet it is not clear whether the actual radius used in the algorithm is small enough such that this assumption holds. Second, the algorithm can only obtain an estimated gradient and using that could lead to very wrong results in computing the optimal tangent direction (as this gradient estimation is well-known to be not precise even using large B). As a motivation for the proposed method, these two assumptions sound ok but the authors might want to further verify the actual conditions in the experiments to make them more convincing.

2. Another downside is that the proposed method will need to tune extra parameters like r compared to other baselines. The authors might also want to show an ablation study on r.

3. In the experiments part, the authors only consider the L2 norm threat model. I think there is no obstacle here to also apply the proposed method in Linf norm settings? (correct me if I miss something here). It would be more comprehensive to also see the Linf norm experiments and compare them with SOTA methods like HSJA/RayS. Especially for the defended models, as they are not trained in the L2 threat model, which makes the results less representative.

4.  Also in the experiments (especially CIFAR10), the advantages of the proposed method are marginal or even non-exist compared with other baselines when the number of queries is sufficient. Here the author only considered 10K as the maximum query number. It is better to show whether the proposed method still holds up the position with maximum of 20k or 50k queries. It would also be nice to have some plots instead of tables to show this one.

=====after rebuttal=========
I thank the authors for their detailed responses. And I appreciate all the extra experiments conducted by the authors. Yet I feel that the proposed method is not better than RayS or QEBA-S on different settings (in R2's comments). And comparing to HSJA, the advantages get vanished when the maximum query number is large or dealing with defended Linf norm models. Therefore, I will keep my score unchanged.


**Time Spent Reviewing:**

4

---

> ### Author Response · Authors · 2021-08-10
> **Response to Reviewer R4**
>
> The authors thank the reviewer for your helpful comments to enhance our paper, and we shall carefully revise the paper following your suggestions.
>
> **Q1**: $H$ can be seen as a hyperplane only when we are considering a very small local region, and whether the actual radius used in the algorithm is small enough such that this assumption holds?
>
> **A1**: The proof of Theorem 1 uses the assumption that $H$ is locally flat, but neither the derivation nor the algorithm needs this assumption to work.
> As shown in the derivation of our algorithm, we only need to focus on the decision boundary restricted to a two-dimensional plane (*e.g.,* Figure 1).
> If the decision boundary is curved "downwardly" (as opposed to the example in Fig. 3b), then searching along the tangent line of the hemisphere is still better than the HSJA solution.
> The worst case is when the boundary is curved "upwardly" and has a large curvature (Fig. 3b), which is handled by the semi-ellipsoid variant of our algorithm.
> We replace the hemisphere with a semi-ellipsoid and use a parameter $r$ to control its shape: the semi-ellipsoid becomes "slender" by setting a large $r$, which increases the height of the tangent point $\mathbf{k}$ to deal with this case.  This is verified in the experiments (*e.g.,* Figure 4).
>
>
> **Q2**: The estimated gradient could lead to very wrong results in computing the optimal tangent direction, as this gradient estimation is not precise.
>
> **A2**:
> (1) It is true that the estimated gradient $\hat{g}$ can be inaccurate, but as long as it does not deviate too much from the true gradient $g$ (*i.e.,* $\langle\hat{g},g\rangle>\eta$ for some constant $\eta>0$), the estimated gradient is enough for our algorithm to make progress. Such estimated gradient is obviously more accurate than pure random directions, which is verified in the experimental result of Figure 5(c) of our paper. Our all experimental results show that Tangent Attack performs well and it stably surpasses HSJA (Table 1 and Table 2 and other figures).
> In addition, the batch size $B_t$ is increased with iterations to increase the accuracy of estimation in the later stage of the iteration.
>
> (2) Such gradient estimation is asymptotically unbiased as $\delta \rightarrow 0^+$, so we have $\lim_{\delta\rightarrow 0} \cos(\mathbb{E}[\tilde{\nabla S}(\mathbf{x}_t,\delta)],\nabla S_\mathbf{x} (\mathbf{x}_t)) = 1$. In particular, the cosine similarity between $\tilde{\nabla S}(\mathbf{x}_t,\delta)$ and true gradient $\nabla S_\mathbf{x} (\mathbf{x}_t)$ is bounded as $\cos(\mathbb{E} [\tilde{\nabla S}(\mathbf{x}_t,\delta)], \nabla S_\mathbf{x} (\mathbf{x}_t)) \geq 1 - \frac{9L^2 \delta^2 d^2}{8\vert\vert\nabla S(\mathbf{x}_t)\vert\vert_2}$. The proof can be found in Appendix A-B of [1]. Also, almost all other black-box attacks use the estimated gradient for attack, so this is a common practice of black-box attacks.
>
> (3) As we have answered in the A1 for the first reviewer R1, we use two techniques to control the estimation error, *i.e.,* the adaptive $\delta$ and the subtraction of the baseline for reducing its variance.
>
>
> **Q3**: The authors might also want to show an ablation study on $r$.
>
> **A3**: Tangent Attack (hemisphere) does not require any additional hyperparameter, and $r$ is used in the semi-ellipsoid variant. We provide the ablation study figures of $r$ in this [downloadable link](https://drive.google.com/file/d/1nE7Hfa2ID-MPK22sM0qI_G6r9MBphG4W/view?usp=sharing), and the performance is not very sensitive to $r$. This figure will be added in the revised version of our paper.
>
> **Q4**: $\ell_\infty$ norm attack experimental results?
>
> **A4**: All the results of $\ell_\infty$ norm attack are presented in Appendix of our paper (Supplementary Material). The result of RayS can be found in the answer A2 to the second reviewer R2. We also offer the $\ell_\infty$ norm experimental results for the defended models as the below table, which will be added to the Appendix.
>
> Defense | Method  | @1K | @2K | @5K | @8K | @10K |
> :- |  :- | :- |:- |:- |:- | :- |
> AT | HSJA | **0.145** | **0.121** | **0.090** | 0.080 | **0.075** |
> AT | Tangent Attack(hemishpere) | 0.147 | **0.121** | **0.090** | **0.079** |**0.075** |
> AT | Tangent Attack(semi-ellipsoid) | 0.149 | 0.123 | 0.091 | 0.080 |0.076 |
> TRADES| HSJA | 0.162 | **0.137** | **0.106** | **0.095** | **0.090** |
> TRADES| Tangent Attack(hemishpere) | 0.163 | 0.138 | 0.107 | **0.095** |**0.090** |
> TRADES| Tangent Attack(semi-ellipsoid) | **0.160** | **0.137** | 0.107 | 0.096 |0.091 |
>
> From the above table, our performance is the comparable to HSJA. Because under the definition of $\ell_\infty$ distance: $D_{\ell_\infty}(x,y):=\max_i(|x_i - y_i|),i\in\\{1,\cdots,dim\\}$, the intersection of the tangent line and the decision boundary may not give the minimum distance to the original image. So searching along the tangent line cannot always outperform HSJA in $\ell_\infty$ norm distortion. In conclusion, our approach performs better on $\ell_2$ norm attacks and is comparable to HSJA on $\ell_\infty$ norm attacks, *e.g.,* Figure 4 in the main text of our paper and Figures 4,5,6 in the Appendix for $\ell_2$ norm results of AT and TRADES.
>
>
> **Q5**: Whether the proposed method still holds up the position with maximum of 20k or 50k queries? Show some plots.
>
> **A5**: The plots of targeted attacks on 5 ImageNet models under the maximum query budget of 20K are presented in [this downloadable link](https://drive.google.com/file/d/1ROETDntt7c1XiMzMdAmsjEETPU09S92Q/view?usp=sharing). When the number of queries is very large (>10K), the performance gap between HSJA and Tangent Attack may become smaller because the obtained information is large enough that both can achieve successful attacks. The extent to which this performance gap is reduced depends on the model being attacked. It should be noted that in real-world scenarios, the number of queries is usually extremely limited, and can never exceed 10K. Therefore, the proposed method is practical and useful in attacking real-world systems.
>
> **Q6**: Societal impact not applicable?
>
> **A6**: As stated in the checklist section, it is presented in the Appendix (supplementary material) of our paper.
>
> [1] Jianbo Chen, Michael I Jordan, and Martin J Wainwright. HopSkipJumpAttack: a query-efficient decision-based adversarial attack. In 2020 IEEE Symposium on Security and Privacy (SP). IEEE, 2020.

---

> ### Author Response · Authors · 2021-08-23
> **The second Response to Reviewer R4**
>
> The authors thank you for the response. We still recommend that you reconsider your decision, because:
>
> **Q7**: The proposed method is not better than RayS or QEBA-S.
>
> **A7**: There is a motto about academic research which makes sense, let us quote it here: "**The academic is not an army race. It does not really matter how fancy the model is. It does not really matter whether the model can achieve the stoa performance. The real innovation is to find something new and this work has found a fresh new perspective.**"
>
> We discover a new geometric-based method to mount the hard-label attacks, this **opens a new gate that makes the community find endless new attack methods from the knowledge of geometry**. As you have mentioned, our approach can improve HSJA consistently across models and datasets, which proves our approach's effectiveness. Importantly, Tangent Attack can further improve the performance of other HSJA-based method (e.g., QEBA-S) when combining with it!
>
> The goals and settings of RayS [1] and QEBA-S [2] are completely different from the proposed approach. The main focus of our study is the targeted attack, and then our method can also be expand to untargeted attacks. So we use an algorithmic process that is **more general rather than specific to one type of attack**.
> RayS is only applicable to untargeted attack under $\ell_\infty$ norm constraint, and it does not even support $\ell_\infty$ norm targeted attack or $\ell_2$ norm attack. QEBA-S [2] only supports the targeted attack.
> In contrast, the proposed method supports all type of attacks under all settings. So it is not fair to compare with them on one specific type of attack.
>
> This is because RayS [1] focuses on finding adversarial example efficiently **only** in the untargeted attack under $\ell_\infty$ norm constraint. RayS starts from the original image and attempts to find all possible directions to reach the untargeted adversarial example.  In untargeted attacks, it is easy because moving the original sample far enough in any direction can make it escape the original true class's area. However, in targeted attack, it is very difficult to find a suitable direction along which can reach the targeted class's region. In contrast, our approach starts from an image of the target class, and then reduces its distortion while staying in adversarial region, thereby gradually making it closer to the original image. Such algorithmic process is applicable to all types of attacks, and the success rate is 100%.
>
> It should be emphasized that our approach can be used as a performance enhancing plug-in when combining with QEBA-S, *e.g.,* QEBA-S + Tangent Attack(hemisphere) surpasses QEBA-S in the below table (targeted attack under $\ell_2$ norm result):
>
> | Method | @1K | @2K | @5K | @8K | @10K |
> | :- | :- |:- |:- |:- | :- |
> | QEBA-S  | 65.163 | 53.075 | 30.057 | **19.292** | 15.469 |
> | HSJA  | 71.340 | 61.553 | 40.311 | 27.905 |  23.305 |
> | Tangent Attack(hemisphere)  | 65.456 | 55.171 | 35.460 | 25.451 | 21.170 |
> | Tangent Attack(semiellipsoid) | 65.819 | 55.611 | 35.596 | 25.286 | 21.063 |
> | QEBA-S + Tangent Attack(hemisphere)  | **62.577** | **50.364** | **28.605** | 19.397 | **15.376** |
>
> In conclusion, the proposed approach can work on the settings that QEBA-S and RayS cannot do, and it supports all types of attacks. Importantly, Tangent Attack can further improve the performance of QEBA-S when combining with it.
>
> **Q8**: Comparing to HSJA, the advantages get vanished when the maximum query number is large.
>
> **A8**: When the number of queries is very large (>10K), the performance gap between HSJA and Tangent Attack may become smaller. This is an inevitable result to some extent, because:
>
> (1) It has reached the late stage of the iteration when the comsuming queries reach 10K~20K, and in late iterations the adversarial example is already very close to the original image. Thus the $\ell_2$ distortion becomes very small and consequently the performance gap between HSJA and Tangent Attack may become smaller.
>
> Also, other HSJA-based methods have exhibited similar result, *e.g.,* The performance of Policy Driven Attack [3] began to lag behind HSJA when the number of queries > 10K (see "@25K" in Tables 1, 2  of their paper[3]).
>
> (2) With 20K queries, the obtained information is large enough that both can achieve successful attacks for most samples. So the performance gap may become smaller.
>
> (3) The [plots](https://drive.google.com/file/d/1ROETDntt7c1XiMzMdAmsjEETPU09S92Q/view?usp=sharing) show that Tangent Attack still has sufficient performance advantages over HSJA in attacking Inception-v3, Inception-v4 and SENet-154 even at 20K maximum queries!
>
> (4) It is almost impossible to find a realistic system that allows 20K queries on only one image, so the number of queries is usually extremely limited. Therefore, the proposed method is practical and useful in attacking real-world systems.
>
> [1] Jinghui Chen and Quanquan Gu. Rays: A ray searching method for hard-label adversarial attack. In Proceedings of the 26rd ACM SIGKDD International Conference on Knowledge Discovery and Data Mining, 2020.
>
> [2] Ali Rahmati, Seyed-Mohsen Moosavi-Dezfooli, Pascal Frossard, and Huaiyu Dai. Geoda: a geometric framework for black-box adversarial attacks. In Proceedings of the IEEE/CVF Conference on Computer Vision and Pattern Recognition, pages 8446–8455, 2020
>
> [3] Ziang Yan, Yiwen Guo, Jian Liang, and Changshui Zhang. Policy-driven attack: Learning to query for hard-label black-box adversarial examples. In International Conference on Learning Representations, 2021.

---

### Official Review · Reviewer_geDq · 2021-07-15

**Rating:** 6
**Confidence:** 2

**Summary:**

This work proposes a new type of hard-label attacking method. Hard-label attacking methods only allow access of the top-1 predicted labels. This method is based on geometric properties, i.e., tangent points to reduce the l2 distortion after attacks. The method is shown to be superior compared with the existing hard-label attacking methods. The authors also studied the performance under four different defense frameworks, all of which prove the superiority of the proposed method.






**Ethical Concerns:**



**Limitations And Societal Impact:**

No analysis has been made.

----
## Post-Rebuttal Update
----

Limitation and Societal Impact are addressed in Appendiex.

**Main Review:**

Main concerns:

1. There is something wrong with the fonts and the styles. The authors should have a look and made corrections.

2.  Although the author claims it is hyperparameter-free, $r$ is still one of the hyper-parameter (although empirically set in the article). If one wants to use the method on other datasets, should the value of $r$ change?


----
## Post-Rebuttal Update
----
I thank the authors for responding to my questions. I will go through others' comments and make discussions but currently, I will keep my score.

**Time Spent Reviewing:**

3

---

> ### Author Response · Authors · 2021-08-10
> **Response to Reviewer R3**
>
> We would like to thank the reviewer for the positive comments, and we shall carefully revise the paper following your suggestions.
>
> **Q1**: There is something wrong with the fonts and the styles.
>
> **A1**: We use a package of "fouriernc" in LaTeX, which is conflict with NeuraIPS's style that makes the font a little different. We will remove the "\usepackage{fouriernc}" statement in the revised version.
>
> **Q2**: Should the radius ratio $r$ changed when using the method on other datasets?
>
> **A2**: Yes, $r$ should be changed a little according to the experimental performance. However, the performance is not very sensitive to $r$. We provide the ablation study figures of $r$ in this [downloadable link](https://drive.google.com/file/d/1nE7Hfa2ID-MPK22sM0qI_G6r9MBphG4W/view?usp=sharing). It should be noted that Tangent Attack (hemisphere) does not require any additional hyperparameter, and $r$ is used in the semi-ellipsoid variant.
>
> **Q3**: No analysis has been made for limitations and societal impact?
>
> **A3**: As stated in the checklist section, they are presented in the Appendix (supplementary material) of our paper.

---

### Official Review · Reviewer_qsse · 2021-07-16

**Rating:** 3
**Confidence:** 5

**Summary:**

The paper proposes Tangent Attack, a decision based adversarial attack which can be seen as an improvement of HopSkipJumpAttack [6]: it progressively reduces the norm of the adversarial perturbations according to a method inspired by the geometry of the decision boundary in the input space. This allows to achieve adversarial points with lower distortion than HSJA with the same amount of queries of the classifiers, i.e. better query efficiency.

**Ethical Concerns:**

No.

**Limitations And Societal Impact:**

I think both are sufficiently discussed.

**Main Review:**

Strengths
- The motivation of the method and the geometric approach are simple and reasonable.

- The improvement upon HSJA is consistent across models and datasets.

Weaknesses
- The main concern is about the experimental evaluation: the authors cite many related works (QEBA, SurFree, GeoDA) which are improvements of HSJA, as acknowledged, but do not include them in the experimental comparison. Since all those methods outperform HSJA, Tangent Attack needs to be compared to them, especially given that its improvement over HSJA is often not large.

- Similarly, for the $\ell_\infty$-threat model RayS should be included.

Other comments
- The template, especially the font, seems to be not the original one.

- The target classes for targeted attacks are usually sampled randomly, not as the next one.

Overall, the method is quite reasonable and well presented, but the experimental evaluation not sufficient.

**Time Spent Reviewing:**

2

---

> ### Author Response · Authors · 2021-08-10
> **Response to Reviewer R2**
>
> The authors thank the reviewer for your helpful comments to enhance our paper, and we shall carefully revise the paper following your suggestions.
>
> **Q1**: Related works (QEBA, SurFree, GeoDA) are not included in the experimental comparison.
>
> **A1**: The main focus of our study is the targeted attack, and then our method can also be used in untargeted attacks. Hence we use an algorithmic process that is **more general rather than specific to one type of attack.** Therefore, for complete and fair comparisons, we select the compared methods that supports **both** untargeted attacks and targeted attacks under **both** $\ell_2$ norm and $\ell_\infty$ norm constraints without requiring auxiliary models, as shown in Table 1 of our paper. SurFree does not support $\ell_\infty$ norm attack and targeted attack. In addition, SurFree is officially published as a CVPR 2021 paper in June 11, 2021, which is late than NeurIPS's submission deadline May 28, 2021.
>
> GeoDA starts from the original image and attempts to find a optimal direction to reach the adversarial example. GeoDA does not support targeted attacks because GeoDA directly shifts the original image along the gradient direction estimated from the previous boundary adversarial sample to find the next adversarial example. Such algorithmic process is not applicable to the targeted attack, because it is very difficult to find a suitable direction along which the targeted class's examples exist.
>
> QEBA reduces the dimension of sampled random perturbations to estimate more effective gradients, which has three variants: QEBA-I, QEBA-S and QEBA-F. QEBA-S use bilinear interpolation operation to resize the low-dimensional perturbations; QEBA-F employs DCT to reduce dimension; and QEBA-I needs to pretrain a surrogate model to calculate gradients for PCA. The original QEBA only supports the targeted attack.
> In contrast, our approach supports all types of attacks with 100% success rate without requiring any surrogate model.
>
> We modify QEBA-S code to make it support untargeted attacks. We present the $\ell_2$ distortion results of GeoDA and QEBA-S by performing untargeted attacks under $\ell_2$ norm constraint on ResNet-101 of the ImageNet dataset in the below table.
>
> | Method | @1K | @2K | @5K | @8K | @10K |
> | :- | :- |:- |:- |:- | :- |
> | Boundary Attack | 64.007 | 60.389 | 56.544 | 44.175 |31.371 |
> | Sign-OPT | 38.282 | 21.985 | 10.048 | 7.050 | 6.050 |
> | SVM-OPT | 32.638 | 19.409 | 9.830 | 7.185 | 6.281 |
> | GeoDA  | 17.968 | 12.526 | 8.151 | 7.133 | 6.364 |
> | QEBA-S| **15.375** | **8.988** | **3.749** | **2.473** | **2.183** |
> | HSJA  | 27.443 | 17.717 | 7.649 | 4.723 |  4.019 |
> | Tangent Attack(hemisphere) | 26.777 | 17.651 | 7.730 | 4.822 | 4.107 |
> | Tangent Attack(semiellipsoid) | 26.631 | 17.384 | 7.602 | 4.720 | 4.026 |
>
> In the above table, QEBA-S and GeoDA surpass our approach. GeoDA is more suitable in untargeted attacks, because it can always find a non-true class's region in any direction far away from the original point in untargeted setting, while in targeted setting, such chances are low. Hence GeoDA does not support targeted attacks, whereas our approach performs well in all types of attacks. This is because our approach starts from an image of the target class, and then reduces its distortion while staying in adversarial region, thereby gradually making it closer to the original image. Such algorithmic process is applicable to all types of attacks, and the success rate is 100%.
>
> QEBA-S performs well in the above table because it reduces dimension in gradient estimation, and one unique benefit of our approach is that it can be used as a performance enhancing plug-in when combining it with other HSJA-based approaches (*e.g.,* QEBA). For example, we present that "QEBA-S + Tangent Attack(hemisphere)" can further improve the performance of QEBA-S in the last table (the table in the answer A4 to the last question).
>
>
> **Q2**: RayS should be included in $\ell_\infty$ norm attack.
>
> **A2**: We did not include RayS in the $\ell_\infty$ norm attack (located in appendix), because RayS only supports untargeted attack under $\ell_\infty$ norm constraint while our approach supports all types of attacks (*e.g.,* targeted one under $\ell_2$ norm).
>
> RayS does not support targeted attacks, because RayS starts from the original image and attempts to find a optimal direction to reach the adversarial example. Rays uses a hierarchical approach to search for all possible directions, and it is very difficult to find a suitable direction along which the targeted class's region exist. Thus RayS does not support targeted attacks for the same reason as GeoDA.
>
> Nonetheless, we will add the RayS results in the revised version of appendix. Here, we show its $\ell_\infty$ norm result of attacking Inception-v3 network on the ImageNet dataset in the below table:
>
> | Method  | @1K | @2K | @5K | @8K | @10K |
> | :- | :- |:- |:- |:- | :- |
> | Sign-OPT  | 0.726 | 0.513 | 0.335 | 0.280 | 0.260 |
> | SVM-OPT | 0.763 | 0.526 | 0.336 | 0.280 | 0.260 |
> | GeoDA  | 0.080 | 0.057 | 0.037 | 0.033 | 0.029 |
> | RayS  | **0.061** | **0.039** | **0.025** | **0.021** | **0.019** |
> | HSJA  | 0.236 | 0.174 | 0.093 | 0.069 |  0.059 |
> | Tangent Attack(hemisphere) | 0.234 | 0.173 | 0.093 | 0.068 | 0.059 |
> | Tangent Attack(semi-ellipsoid) | 0.237 | 0.174 | 0.093 | 0.069 | 0.061 |
>
> In the above table, RayS performs well because it uses a hierarchical approach and a fast check step to skip unnecessary searches. However, it is difficult find a suitable direction along which the original image can reach the target class's adversarial region. Furthermore, RayS does not support $\ell_2$ norm attack because its discrete formulation is defined based on $\ell_\infty$ norm, whereas our approach performs well on all types of attacks (*e.g.,* targeted one under $\ell_2$ norm).
>
> **Q3**: The font seems to be not the original one.
>
> **A3**: We use a package of "fouriernc" in LaTeX, which is conflict with NeuraIPS's style that makes the font a little different. We will remove the "\usepackage{fouriernc}" statement in the revised version.
>
> **Q4**: The target classes for targeted attacks are sampled randomly, not as the next one.
>
> **A4**: The results of targeted attack under $\ell_2$ norm with random target classes on ResNet-101 of the ImageNet dataset are shown in the below table.
>
> | Method | @1K | @2K | @5K | @8K | @10K |
> | :- | :- |:- |:- |:- | :- |
> | QEBA-S  | 65.163 | 53.075 | 30.057 | **19.292** | 15.469 |
> | HSJA  | 71.340 | 61.553 | 40.311 | 27.905 |  23.305 |
> | Tangent Attack(hemisphere)  | 65.456 | 55.171 | 35.460 | 25.451 | 21.170 |
> | Tangent Attack(semiellipsoid) | 65.819 | 55.611 | 35.596 | 25.286 | 21.063 |
> | QEBA-S + Tangent Attack(hemisphere)  | **62.577** | **50.364** | **28.605** | 19.397 | **15.376** |
>
> As we can see from above table, although QEBA-S slightly surpasses our approach, one unique benefit of Tangent Attack is that it can be used as a performance enhancing plug-in when combining with other HSJA-based approaches, *i.e.,* QEBA-S + Tangent Attack(hemisphere) surpasses QEBA-S. We will explore more such examples in the future work.
>
> **Last words:**
> Finally, there is a motto about academic research which makes sense, let us quote it here: "**The academic is not an army race. It does not really matter how fancy the model is. It does not really matter whether the model can achieve the stoa performance. The real innovation is to find something new and this work has found a fresh new perspective.**"
>
> We discover a new geometric-based method to mount the hard-label attacks, this **opens a new gate that makes the community find endless new attack methods from the knowledge of geometry**.  As you have mentioned, our approach can improve HSJA consistently across models and datasets, which proves our approach's effectiveness.
> Importantly, Tangent Attack can further improve the performance of other HSJA-based method (*e.g.,* QEBA-S) when combining with it!

---

> > ### Comment · Reviewer_qsse · 2021-08-23
> > **Post rebuttal comments**
> >
> > I thank the authors for the response and the additional experiments.
> >
> > I agree that the flexibility of handling different norms and both targeted and untargeted attacks is a strength of the proposed method, but I think this should be discussed in details in the paper and is not sufficient to justify the exclusion of the SOTA attacks from the comparison. Also, I'm not sure that GeoDA and RayS can't be used in the targeted scenario, since a valid starting point, classified in the target class, can be found from an image e.g. from the training set belonging to the target class (see e.g. https://arxiv.org/abs/1903.10826).
> > Moreover, in the reported results, QEBA-S and RayS outperform Tangent Attack and this achieves similar results to HSJA for large numbers of queries.
> > Finally, the fact that Tangent Attack can be combined with existing methods is interesting but should be discussed in the paper with the support of experimental results.
> >
> > Overall, I don't think the additional results support the proposed method, then I keep my original score.

---

> > > ### Author Response · Authors · 2021-08-23
> > > **The second response to Reviewer R2**
> > >
> > > The authors thank you for the response.  However, we have a statement to say at first, and then we will answer your questions.
> > >
> > > There is a motto about academic research which makes sense, let us quote it here: "**The academic is not an army race. It does not really matter how fancy the model is. It does not really matter whether the model can achieve the stoa performance. The real innovation is to find something new and this work has found a fresh new perspective.**"
> > >
> > > We discover a new geometric-based method to mount the hard-label attacks, this **opens a new gate that makes the community find endless new attack methods from the knowledge of geometry**.  As you have mentioned, our approach can improve HSJA consistently across models and datasets, which proves our approach's effectiveness.
> > > Importantly, Tangent Attack can further improve the performance of other HSJA-based method (*e.g.,* QEBA-S) when combining with it!
> > >
> > > The goals and settings of your mentioned method (RayS and GeoDA) are completely different from the proposed approach.  The main focus of our study is the targeted attack, and then our method can also be expand to untargeted attacks. However, based on the reason we mentioned these attacks cannot be applicable to the targeted attack, and we use an algorithmic process that is **more general rather than specific to one type of attack.** So it is not fair to compare with them in such a specific type of attack.  In conclusion, the proposed approach can work on the settings that QEBA-S,GeoDA and RayS cannot do, and it supports all types of attacks.
> > >
> > > **Q5**: The flexibility of handling different norms and both targeted and untargeted attacks should be discussed in details in the paper.
> > >
> > > **A5**: We will revise the following part or appendix of our paper to discuss it and justify the exclusion of your mentioned attacks, as we have answered in A1.
> > >
> > > **Q6**: I'm not sure that GeoDA and RayS can't be used in the targeted scenario, but the method of https://arxiv.org/abs/1903.10826 can?
> > >
> > > **A6**: GeoDA [1] and RayS [2] do not support targeted attacks. The "qFool" method in your provided link (https://arxiv.org/abs/1903.10826) **starts from an image of the target class** (Section 4 of their paper), and then reduces its distortion while staying in adversarial region, thereby gradually making it closer to the original image. Such process is similar to HSJA and Tangent Attack (see Figure 2 of qFool paper), and which is completely different from GeoDA and RayS. Both GeoDA [1] and RayS [2] **start from the original clean image** and find a possible direction to reach the adversarial example.  In untargeted attacks, it is easy because moving the original sample far enough in any direction can make it escape the original true class's area. However, in targeted attack, it is hard to find a suitable angle along which can reach the targeted class's image, as we have mentioned. In addition, the author of RayS confirmed this fact in the GitHub issue of their official code repository.
> > >
> > > By the way, the papers of GeoDA [1] and RayS [2] do not provide any experiment of the targeted attack.
> > >
> > > **Q7**: The fact that Tangent Attack can be combined with existing methods is interesting, but should be discussed in the paper with the support of experimental results.
> > >
> > > **A7**: We will add the discussion and the experimental results of the combination "QEBA-S + Tangent Attack" (which has been presented in the answer A4) into the revised version of our paper.
> > >
> > > [1] Ali Rahmati, Seyed-Mohsen Moosavi-Dezfooli, Pascal Frossard, and Huaiyu Dai. Geoda: a geometric framework for black-box adversarial attacks. In Proceedings of the IEEE/CVF Conference on Computer Vision and Pattern Recognition, pages 8446–8455, 2020.
> > >
> > > [2] Jinghui Chen and Quanquan Gu. Rays: A ray searching method for hard-label adversarial attack. In Proceedings of the 26rd ACM SIGKDD International Conference on Knowledge Discovery and Data Mining, 2020.

---

### Official Review · Reviewer_qy3F · 2021-07-21

**Rating:** 7
**Confidence:** 3

**Summary:**

This paper proposes a new method for performing hard-label black-box attacks, similar to the HopSkipJump Attack but modified with the observation that the gradient direction is not always the best direction to move in when trying to minimize mean attack distortion. Instead, under some assumptions about the decision boundary, the paper derives an update direction based on an optimal tangent point on the semi-circle around the current attack iterate. The paper then implements the corresponding algorithm and shows that it indeed lowers mean distortion on the CIFAR-10 and ImageNet datasets at a variety of query budgets.

**Limitations And Societal Impact:**

Yes, the authors dedicate an entire appendix section to discussing social impact, and also have a detailed limitations section (in the Appendix). If space permits it would be nice to see some of these discussed in the main paper.

**Main Review:**

The proposed method seems reasonable and to my knowledge the given proofs are correct. I am not an expert in geometry, however, so I can't judge the technical depth or novelty of the theoretical contributions. On the empirical side, the evaluation seems solid (modulo some concerns listed below) and the method does improve on competing hard-label attack algorithms, which lends some credence to the approach. My main comments and questions are below:

Presentation/Clarity:

- L48: The paper claims that "although Rahmati et al. claimed that GeoDA is grounded on the geometric properties of DNNs, it assumes that the decision boundary of DNNs is flat in the vicinity of data samples"---however, isn't this flatness assumption also made by the proposed hemisphere attack as well? I think the authors should either better distinguish their method from GeoDA or just remove the corresponding claim from the paper.
- Fig. 1: What is the motivation for the oddly-shaped decision boundary? In particular, what purpose does the upper decision boundary line serve? In my opinion this figure should be as simple as possible since it is the reader's first introduction to the main idea, so any lines that can be removed should be removed.
- Proof of Theorem 1: Any assumptions made in the proof of Theorem 1 (in particular, Assumption B.1 in the appendix) should be stated in the main text (and explained in intuitive terms if possible) rather than in the proof itself.
- I thought the layout of the paper was good, but local readability is hindered a bit by some grammatical/spelling/phrasing issues (e.g., missing articles, missing pluralizations, tense agreement, etc). These did not affect my score but I think the clarity of the paper could be improved with some proofreading.
- The legend of Figure 5 is too small/hard to read
- Again minor and did not affect my score, but the entire paper is typeset in an incorrect font (you probably have a stray \usepackage in the LaTeX source).


Experimental:
- Rather than just the average l2 distortion, I think the paper should also show the distribution of l2 distortions across test images---i.e., is it the case that the l2 distortions found by the semi-circle attack are uniformly better, or is it just the other attacks "fail" on some images leading to very high mean distortions?
- The paper mentions that HSJA was re-implemented in PyTorch for consistency which is reasonable, but as a sanity check the appendix should include a comparison of the re-implemented HSJA and the original HSJA codebase on the same models (since it's a black-box attack, it should be easy to use a PyTorch model [or any arbitrary function] in fact) in conjunction with the original code.
- I like the ablation studies that the authors performed in Figure 5! In Figure 5c it seems like decreasing the batch size monotonically improves the performance of the attack, up to the smallest shown batch size. Did the authors try lowering the batch size even further, e.g., to 15 or 10? It would be interesting to see at what point the stability from batching outweighs the redundant queries.


**Time Spent Reviewing:**

4

---

> ### Author Response · Authors · 2021-08-10
> **Response to Reviewer R1**
>
> We would like to thank the reviewer for the positive comments, and we shall carefully revise the paper following your suggestions.
>
> **Q1**: It wrote "GeoDA heavily relies on the assumption of flat decision boundary to estimate the gradient". Isn't this flatness assumption also made by the proposed hemisphere attack as well?
>
> **A1**: Both GeoDA and our approach sample $B$ random vectors to estimate the gradient. In GeoDA, the gradient is estimated as $\hat{g} = \frac{1}{B} \sum_{i=1}^B \rho_i \cdot \mathbf{u}_i$, where $\mathbf{u}_i \sim \mathcal{N}(\mathbf{0},\mathbf{1})$ and $\rho_i$ marks whether a perturbed sample resides in the adversarial region, *i.e.,* $\rho_i=1$ if $\arg\max_k f({\mathbf{x}_t + \delta \mathbf{u}_i})_k \neq \arg\max_k f(\mathbf{x})_k$, otherwise $\rho_i=-1$. The parameter $\delta$ is a small constant to control the estimation precision, which is set to 0.0002 in all GeoDA's experiments. In the situation of a large curved decision boundary, there is an uneven distribution of perturbed samples at the two sides of the boundary. A small fixed $\delta$ causes nearly all $\mathbf{x}_t + \delta \mathbf{u}_i$ to locate on the same side of boundary, thereby resulting in the inaccurate estimation. Thus, GeoDA only works on classifiers with low-mean-curvature or flat boundary, as stated in their paper.
>
> Our approach uses the following formular to estimate gradient:
>
> $$\tilde{g} = \frac{1}{B-1} \sum_{i=1}^{B} \left(\phi(\mathbf{x}_t + \delta \mathbf{u}_i) - \tilde{\phi} \right) \mathbf{u}_i $$
>
> where $\tilde{\phi} = \frac{1}{B} \sum_{i=1}^{B} \phi(\mathbf{x}_t + \delta \mathbf{u}_i)$ is the baseline and $\phi(\mathbf{x}_t + \delta \mathbf{u}_i) = \rho_i$ marks a successful attack with $\mathbf{x}_t + \delta \mathbf{u}_i$. It uses two techniques to control the estimation error: (1) the $\delta$ is set adaptively; (2) it subtracts the baseline to reduce the variance when $\mathbb{E}[\phi(\mathbf{x}_t + \delta \mathbf{u})] > 0$, which can reduce the estimation error caused by the uneven distribution of perturbed samples. More details can be found in [1]. Therefore, the proposed method can estimate a relatively more accurate gradient in the curved decision boundary.
>
>
> **Q2**: What is the motivation for the oddly-shaped upper decision boundary in Fig. 1?
>
> **A2**: If there is no such upper decision boundary, there is no limitation to the step size of the HSJA method along the gradient direction. If this happens, HSJA may jump to a higher position to obtain the same optimal boundary point $\mathbf{x}_t$ as the Tangent Attack. However, in real scenarios, the adversarial region is surrounded by decision boundary, so there exists the upper decision boundary that limits the jump height of HSJA. This is why HSJA uses the "geometric progression" method to repeatedly halve the step size to find the upper decision boundary [1]. This is the limitation of HSJA, and our approach compute the tangent point as the optimal update under such restriction.
>
> **Q3**: Any assumptions made in the proof of Theorem 1 should be stated in the main text rather than in the proof itself.
>
> **A3**: This "Assumption" is not really an assumption: it is an invariant ensured by our algorithm. By repeatedly shrinking the radius R (until it reaches some preset lower bound), the algorithm guarantees that Assumption B.1 always holds when we compute the tangent line. This "Assumption" essentially means that there exists a tangent point on the hemisphere *of the adversarial region*, i.e., the feasible region of the optimization problem Eq. (3) is non-empty. We will revise the corresponding part to make it clearer.
>
> **Q4**: The legend of Figure 5 is too small/hard to read.
>
> **A4**: We will enlarge the legend of Figure 5 in the revised version.
>
> **Q5**: Again minor and did not affect my score, but the entire paper is typeset in an incorrect font.
>
> **A5**: We use a package of "fouriernc" in LaTeX, which is conflict with NeuraIPS's style that makes the font a little different. We will remove the "\usepackage{fouriernc}" statement in the revised version.
>
> **Q6**: Is the $\ell_2$ distortion found by Tangent Attack uniformly better then baseline? The paper should also show the distribution of $\ell_2$ distortions across test images.
>
> **A6**: Yes, the $\ell_2$ distortion found by Tangent Attack is uniformly better than baseline. The figures of distribution of $\ell_2$ distortions across test images can be downloaded in [this link](https://drive.google.com/file/d/1GgBOVrZQ69Dlw0mzSChkCWe4nTmFmOyT/view?usp=sharing).
>
> **Q7**: Is the re-implemented of PyTorch version of HSJA consistent with the original one?
>
> **A7**: The original HSJA is implemented by NumPy. We translate it to the PyTorch version by replace each NumPy function to the corresponding PyTorch function. In addition, we directly use the original NumPy function to generate random vectors to avoid potential inconsistencies in the behavior of random generators. So the behavior of the PyTorch version of HSJA is exactly the same with the original one.
>
> **Q8**: Did the authors try lowering the batch size even further (e.g., to 15 or 10) in Figure 5?
>
> **A8**: The effects of the batch size of 5 and 10 are shown in [this link](https://drive.google.com/file/d/1SW7V5Fg0Pj3lWP6xKlBspmfF-U_DRQrA/view?usp=sharing). In untargeted attacks, the initial batch size $B_0$ of 30 performs the best. In targeted attacks, $B_0=10$ or $B_0=30$ is the best, and $B_0=5$ performs worse than $B_0=30$ because it leads too few probes for the inaccurate gradient estimation.
>
> [1] Jianbo Chen, Michael I Jordan, and Martin J Wainwright. HopSkipJumpAttack: a query-efficient decision-based adversarial attack. In 2020 IEEE Symposium on Security and Privacy (SP). IEEE, 2020.

---

### Decision · Program_Chairs · 2021-09-27

**Decision:**

Accept (Poster)

**Comment:**

The paper proposes a slightly novel approach for black-box adversarial attacks, with extensive sets of experiments that seem to confirm some performance benefits compared to the related work. This one should be better positioned (e.g., targeted vs non-targeted attack settings), and the active discussion with the reviewers should help in clarifying the true benefits of the proposal in the revision of the manuscript towards any eventual publication.